# Sensitivities of modelled water vapour in the lower stratosphere: temperature uncertainty, effects of horizontal transport and small-scale mixing

Liubov Poshyvailo, Rolf Müller, Paul Konopka, Gebhard Günther, Martin Riese, Aurélien Podglajen, and Felix Ploeger

Institute of Energy and Climate Research: Stratosphere (IEK-7), Forschungszentrum Jülich, Jülich, Germany.

*Correspondence to:* Liubov Poshyvailo (l.poshyvailo@fz-juelich.de)

**Abstract.** Water vapour ($H_2O$) in the upper troposphere and lower stratosphere (UTLS) is a key player for global radiation. A realistic representation of $H_2O$ is critical for climate model predictions of future climate change. Here, we investigate the effects of current uncertainties in tropopause temperature, horizontal transport and small-scale mixing on simulated $H_2O$ in the lower stratosphere (LS).

5  To assess the sensitivities of simulated $H_2O$, we use the Chemical Lagrangian Model of the Stratosphere (CLaMS). First, we examine CLaMS driven by two different reanalysis, ERA-Interim and Japanese 55-year (JRA-55) reanalysis, to investigate the robustness with respect to the meteorological dataset. Second, we carry out CLaMS simulations with transport barriers along latitude circles (at the equator, 15° N/S and 35° N/S) to assess the effects of horizontal transport. Third, we vary the strength of parametrized small-scale mixing in CLaMS.

10  Our results show significant differences (about 0.5 ppmv) in simulated stratospheric $H_2O$ due to uncertainties in the tropical tropopause temperatures between the two reanalysis datasets, JRA-55 and ERA-Interim. The JRA-55 based simulation is significantly moister when compared to ERA-Interim, due to a warmer tropical tropopause (approximately 2 K). The transport barrier experiments demonstrate that the Northern Hemisphere (NH) subtropics have a strong moistening effect on global stratospheric $H_2O$. Comparison of tropical entry $H_2O$ from the sensitivity 15° N/S barrier simulation and the reference case shows differences of up to around 1 ppmv. Interhemispheric exchange shows only a very weak effect on stratospheric $H_2O$. Small-scale mixing mainly increases troposphere-stratosphere exchange, causing an enhancement of stratospheric $H_2O$, particularly along the subtropical jets in the summer hemisphere and in the NH monsoon regions. In particular, the Asian and American monsoon systems during boreal summer turn out as regions especially sensitive to changes in small-scale mixing, which appears crucial for controlling the moisture anomalies in the monsoon UTLS. For the sensitivity simulation with varied 20  mixing strength differences in tropical entry $H_2O$ between the weak and strong mixing cases amount to about 1 ppmv, with small-scale mixing enhancing $H_2O$ in the LS.

The sensitivity studies presented here provide new insights into the leading processes that control stratospheric $H_2O$, important for assessing and improving climate model projections.

# 1 Introduction

Stratospheric water vapour ($H_2O$) is a crucial factor for global radiation, as it cools the stratosphere and warms the troposphere (e.g., Forster and Shine, 1999, 2002; Shindell, 2001; Nowack et al., 2015). Particularly, changes in $H_2O$ mixing ratios in the upper troposphere and lower stratosphere (UTLS) may have significant effects on climate variability (Solomon et al., 2010; Riese et al., 2012; Maycock et al., 2013; Nowack et al., 2017). Thus, the reliability of climate model predictions is significantly affected by the representation of the processes controlling the distribution of stratospheric $H_2O$. However, there is a multitude of such critical processes, poorly understood and quantified hitherto, rendering the representation of stratospheric $H_2O$ a major uncertainty factor for global climate models (Gettelman et al., 2010; Randel and Jensen, 2013).

A critical region for the control of $H_2O$ entering the stratosphere is the tropical tropopause layer (TTL) (Fueglistaler et al., 2009), extending from the level of main convective outflow around $12\,km$ (about $340\,K$ potential temperature) up to altitudes around $18\,km$ (the highest level convection may reach). The TTL has physical and chemical characteristics midway between the troposphere and stratosphere. Because the TTL is a region of mean upward transport it acts as a "gate to the stratosphere" for trace species and pollution with sources in the troposphere.

Transport processes in the TTL are rather complex, involving large-scale upwelling and horizontal advection linked to the residual mean mass circulation, but also large-scale horizontal and small-scale vertical mixing processes. These mixing processes are particularly important during boreal summer, when mass transport related to the residual circulation is weak. Vertical mixing has been shown to affect trace gas transport in the tropical LS (e.g., Mote et al., 1998; Glanville and Birner, 2017). Horizontal transport between the TTL and middle latitudes is strongly influenced by the Asian monsoon anticyclone and other subtropical circulation systems (e.g., Bannister et al., 2004; James et al., 2008; Wright et al., 2011; Randel and Jensen, 2013). Rapid transport from the tropics to middle latitudes occurs mostly above the subtropical jets within the "tropically controlled transition region" (Rosenlof et al., 1997).

Related to the mean upward transport, the TTL includes the region of very low temperatures around the cold-point tropopause, where the moist tropospheric air is freeze-dried to stratospheric values (Brewer, 1949). Thus, the tropical cold-point temperatures control the amount of $H_2O$, which enters the stratosphere (e.g., Wang et al., 2015; Kim and Alexander, 2015). The dehydration occurs as a result of the slow upward and large-scale horizontal motion of air in this region (Holton and Gettelman, 2001), where the nucleation and sedimentation of ice crystals take place, which in essence is a microphysical process controlled by TTL temperatures. The freezing is sensitive not just to large-scale TTL temperatures, but also to microphysical processes controlling the ice crystal number densities, particle size distribution, and fall speed. Several studies focused on the modelling of the detailed cloud microphysical processes (e.g., Jensen and Pfister, 2004; Jensen et al., 2005, 2012). Other recent papers have examined the effect of cloud microphysical processes on the humidity of the TTL and stratosphere using cloud models of varying complexity (e.g., Ueyama et al., 2015; Schoeberl et al., 2014). The tropical stratospheric entry $H_2O$ mixing ratios can be well simulated by the advection through the large-scale temperature field and instantaneous freezing, often described as the "advection-condensation" paradigm (Pierrehumbert and Rocca, 1998; Fueglistaler and Haynes, 2005). However, based on

trajectory studies driven by ECMWF reanalysis, Liu et al. (2011) showed that such results are sensitive to the temperature and vertical velocity fields.

Sublimation of ice, injected by deep convection, has also been argued to be an important factor for the $H_2O$ budget of the tropical LS (e.g., Avery et al., 2017; Jensen and Pfister, 2004). Convection affects the transport of water and ice and influences the temperatures over the convective region, in turn affecting dehydration (e.g., Fueglistaler et al., 2009). The predominant impact of convection has been shown to moisten the TTL by up to 0.7 ppmv at 100 hPa level and even more below this level (e.g., Ueyama et al., 2014, 2015). Similarly, Schoeberl et al. (2014) argued that an increase of convection will increase stratospheric $H_2O$ and tropical cirrus around the cold-point tropopause. At higher levels in the TTL, however, the moistening effect of convection appears very weak (e.g., Schiller et al., 2009).

Above the TTL, $H_2O$ behaves mainly as a tracer, and the tape recorder signal imprinted at the cold-point tropopause ascends deep into the tropical stratosphere (Mote et al., 1996). At higher altitudes in the stratosphere, methane oxidation results in a chemical source for stratospheric $H_2O$ (e.g., LeTexier et al., 1988; Rohs et al., 2006). As a net result of this oxidation process each methane molecule is converted into approximately two $H_2O$ molecules. Hence, the total water vapour (TWV), $TWV = 2CH_4 + H_2O$, is unchanged by transport in the stratosphere and can be regarded approximately constant (e.g., Dessler et al., 1994; Mote et al., 1998; Randel et al., 1998). Therefore, the sum $2CH_4 + H_2O$ is an important value to indicate the amount of water entering the stratosphere (e.g., Kämpfer, 2013).

The annual cycle of TTL temperatures (minimum in boreal winter, maximum in summer) is imprinted on $H_2O$ mixing ratios entering the stratosphere, forming the so-called "tape recorder" signal (Mote et al., 1995, 1996). The summer maximum of tropical $H_2O$ mixing ratios has been argued to be also related, to some degree, to the subtropical monsoon circulations like the Asian monsoon. However, the strength of this effect and the detailed processes involved (e.g., deep convection, large-scale upwelling) is a matter of debate. Furthermore, it has been pointed out that the coupling between ozone, the tropospheric circulation, and climate variability plays an important role in climate change (Nowack et al., 2017). Recent studies have shown that stratospheric ozone changes may cause an increase in global mean surface warming, mostly induced by changes in long-wave radiative feedbacks due to the tropical LS ozone and related stratospheric $H_2O$ and cirrus cloud changes (e.g., Nowack et al., 2015; Dietmüller et al., 2014). Seasonal variations of LS ozone lead to a magnification of the seasonal temperature cycle in the tropics (Fueglistaler et al., 2011). Investigation of these additional effects of stratospheric ozone is an important topic of future research focussed on stratospheric $H_2O$ feedbacks.

Satellite observations suggest that horizontal transport from low latitudes affects the $H_2O$ distribution in middle and high latitudes (Rosenlof et al., 1997; Pan et al., 1997; Randel et al., 2001). Additionally, model simulations confirmed that almost the entire annual cycle of $H_2O$ mixing ratios in the Northern Hemisphere (NH) extratropical LS above about 360 K, with maximum mixing ratios during summer and fall, is caused by horizontal transport from low latitudes (Ploeger et al., 2013). In the respective model, highest $H_2O$ mixing ratios in this region are clearly linked to horizontal transport from low latitudes, mainly from the Asian monsoon.

Based on model simulations, Riese et al. (2012) have shown that little changes in small-scale mixing, which may be related to deformations in the large-scale flow, can cause strong effects on the $H_2O$ distribution in the LS. Consequently, uncertainties

| Simulation type | Abbreviation | Reanalysis dataset | Latitude barriers | Lyapunov exponent ($\lambda_c$, day$^{-1}$) |
|---|---|---|---|---|
| Reference | REF | ERA-Interim | – | 1.5 |
| Reanalysis uncertainty | JRA-55 | JRA-55 | – | 1.5 |
| Horizontal transport effects | BAR-0 | ERA-Interim | 0° | 1.5 |
| | BAR-15 | ERA-Interim | 15° N/S | 1.5 |
| | BAR-15S | ERA-Interim | 15° S | 1.5 |
| | BAR-15N | ERA-Interim | 15° N | 1.5 |
| | BAR-35 | ERA-Interim | 35° N/S | 1.5 |
| Small-scale mixing effects | MIX-no | ERA-Interim | – | ∞ |
| | MIX-weak | ERA-Interim | – | 2.0 |
| | MIX-strong | ERA-Interim | – | 1.0 |

**Table 1.** CLaMS sensitivity simulations with respect to the used reanalysis datasets, horizontal transport barriers and small-scale mixing strengths. Note, that the barriers are 10° in-width with the central latitude indicated in the Table.

in the representation of small-scale characteristics of transport in the LS in models may cause substantial uncertainties in the stratospheric $H_2O$ distribution. This, in turn causes uncertainties in the simulated radiative effect of $H_2O$ and of surface temperatures.

In summary, stratospheric $H_2O$ mixing ratios are a result of the interplay of a multitude of complex processes. As these various processes are influenced by climate change in different ways, long-term changes of stratospheric $H_2O$ are complicated to interpret (e.g., Hegglin et al., 2014) and to predict (e.g., Gettelman et al., 2010). In this paper, we investigate uncertainties of modelling $H_2O$ in the LS with respect to two meteorological datasets, ERA-Interim and JRA-55 (e.g., Dee et al., 2011; Kobayashi et al., 2015; Manney et al., 2017; Davis et al., 2017; Manney and Hegglin, 2018), used to drive transport and freeze-drying, horizontal transport between tropics and extratropics, and small-scale mixing in the Chemical Lagrangian Model of the Stratosphere (CLaMS). For that reason, we carried out a number of sensitivity simulations with CLaMS (see Table 1). Our main results show a significant uncertainty for modelling stratospheric $H_2O$ with respect to the underlying meteorological data (in particular TTL temperatures), even when the most current reanalysis products are used. Furthermore, we find a substantial effect of horizontal transport to moisten the tropical LS and to dry the extratropics. The NH subtropics turn out to be a major moisture source region for the global stratosphere. Finally, small-scale mixing has a strong effect on stratospheric $H_2O$, by increasing diffusive cross-tropopause moisture transport and horizontal mixing in the stratosphere.

The model and datasets which were used, as well as the various sensitivity simulations, are described in Section 2. The results regarding different reanalysis, horizontal transport and small-scale mixing strengths are presented in Section 3. A discussion of the results is presented in Section 4.

## 2 Method

### 2.1 The CLaMS model and simulation set-up

We carried out a number of sensitivity simulations using the Chemical Lagrangian Model CLaMS (McKenna et al., 2002a, b) in its 3D-version (Konopka et al., 2004). CLaMS is a Lagrangian transport model based on 3D-forward trajectories and an additional parametrization of small-scale mixing. The time-dependent irregular model grid is defined by Lagrangian air parcels, which follow the flow. An advantage of the Lagrangian approach for simulating stratospheric transport is the ability to resolve small-scale features, which are often below the possible resolution of high-resolved Eulerian models (McKenna et al., 2002b). Such small-scale features are frequently observed in stratospheric trace gas distributions as elongated filaments, related to the stretching and differential advection in sheared flows (Orsolini et al., 1998).

The advection of forward trajectories in CLaMS is calculated based on a fourth-order Runge-Kutta scheme, as described by McKenna et al. (2002a), using 6-hourly wind fields from meteorological reanalysis data. For vertical transport, CLaMS uses a hybrid vertical coordinate, which is an orography-following $\sigma$-coordinate at the ground and transforming into potential temperature above (Mahowald et al., 2002; Pommrich et al., 2014). Above $\sigma = 0.3$ (about 300 hPa), the vertical coordinate is purely isentropic, and vertical transport is driven by the reanalysis total diabatic heating rate (Ploeger et al., 2010). For the simulations considered here we use a horizontal resolution of about 100 km. The vertical resolution is defined via a critical aspect ratio $\alpha$ of 250 (Haynes and Anglade, 1997). This value expresses the ratio between horizontal and vertical scales, and is about 400 m around the tropical tropopause, degrading below and above it (Konopka et al., 2012). The simulations cover the atmosphere from the surface to about the stratopause, and the number of air parcels advected in the simulations is about 2 millions at each time step.

The parametrization of small-scale mixing in CLaMS is based on the deformation rate in the large-scale flow. Hence, air parcels may be merged, or new air parcels may be inserted at each time step (every 24 h), depending on the critical distances between them. The strength of parametrized small-scale mixing can be controlled by choice of a critical finite-time Lyapunov exponent ($\lambda_c$) which, in turn, determines the critical distances between air parcels (for details see McKenna et al., 2002a; Konopka et al., 2004). Whenever nearest neighbour air parcels move closer than a critical distance during one advection time step, they are merged into a single parcel. Whenever they become further separated than a critical distance, a new air parcel is inserted in between (see McKenna et al., 2002a).

A validation of the CLaMS mixing scheme was presented by Konopka et al. (2005a) in comparison to CRISTA-1 observations. Importantly, the CLaMS mixing parametrization affects both vertical and horizontal diffusivity. Horizontal diffusivity is largely associated with deformation in the horizontal flow. The vertical mixing is mainly related to the vertical shear (Konopka et al., 2004, 2005b).

Stratospheric $H_2O$ in CLaMS is calculated using the CLaMS cirrus module. It includes freeze-drying in regions of cold temperatures, which mainly occurs around the tropical tropopause (dehydration). This, in turn, causes formation and sedimentation of ice particles. The lower boundary for $H_2O$ in CLaMS is taken from reanalysis (ERA-Interim or JRA-55) specific humidity below about 500 hPa. If saturation along a CLaMS air parcel trajectory exceeds a critical saturation (100% with re-

spect to ice), then the H$_2$O amount in excess is instantaneously transformed to the ice phase and partly sediments out. Such simple parametrisation has been adopted in several global Lagrangian studies (e.g., Kremser et al., 2009; Stenke et al., 2008). The saturation mixing ratio is calculated as $\chi_{H_2O} = p_s/p$ for each air parcel trajectory, with the saturation pressure given by $p_s = 10^{-2663.5/T+12.537}$ (Marti and Mauersberger, 1993), where $p$ is the ambient pressure (e.g., Kremser et al., 2009).

For sedimentation, a parametrization is based on a mean ice particle radius, a characteristic sedimentation length and the corresponding fall speed. When the fallen path of the ice particles is calculated from the fall speed and the computation time step $\Delta t$, it is compared with a characteristic sedimentation length of about the vertical grid size (here $l_c = 300\,\mathrm{m}$) which has been empirically optimized by comparison with observations (Ploeger et al., 2013). After this step a respective fraction of ice will be removed. If the parcel is sub-saturated and ice exists, this ice is instantaneously evaporated to maintain saturation.

In addition, methane oxidation is included as a source of H$_2$O in the middle and upper stratosphere. Therefore, hydroxyl, atomic oxygen, and chlorine radicals are taken from a model climatology (for details see Pommrich et al., 2014).

Note, that the CLaMS H$_2$O calculation gives meaningful results only above the tropopause due to the simple parametrization of ice microphysics and not including a convection parametrization. In the stratosphere, however, CLaMS H$_2$O has been shown to agree well with the observations (e.g., Ploeger et al., 2013).

To study the sensitivity of simulated stratospheric H$_2$O regarding different reanalysis temperatures, horizontal transport effects and small-scale mixing, we carried out several CLaMS simulations. As a reference, we consider the run driven by ERA-Interim reanalysis data (Dee et al., 2011). To reach steady state we use a perpetuum technique, where the one year run (for 2011 conditions) is repeated several times. The initial values for the tracer fields at the first day of the simulation are taken from a long-term CLaMS simulation (Pommrich et al., 2014). After one year of the perpetuum calculation, tracer mixing 20  ratios from the 31st of December 2011 are interpolated to the air parcel positions on 1st of January 2011, and the calculation is repeated for 2011 again. After the fourth year, the maximum relative change of H$_2$O mixing ratios between further years of the simulation is very small with the defined resolution and the time step (maximum year to year changes are below 1.0 %). Thus, we use the fifth year of the perpetuum simulation for our further analysis. Restricting the analysis to a single year, instead of calculating a multi-year climatology, has no effect on our conclusions regarding the differences between different simulations, 25  as shown in Appendix A.

First, to assess the robustness of simulated H$_2$O with respect to the meteorological datasets we carry out another CLaMS simulation driven by the Japanese 55-year reanalysis data (JRA-55) (Kobayashi et al., 2015), and compare it to the ERA-Interim based reference simulation. Second, to assess the effects of horizontal transport, we carry out sensitivity simulations with horizontal transport barriers along latitude circles at the equator, at 15°N/S and at 35°N/S (Ploeger et al., 2013). The 30  transport barriers are defined in the model and centred at the given latitude. Their thickness is 10° in latitude (to inhibit diffusive mixing transport), and the barriers extend from the ground to 600 K potential temperature. The two types of barriers, BAR-15 and BAR-35, are located at the edge of the subtropics. BAR-15 is located at the equatorward edge and BAR-35 at the poleward edge of the subtropics. So, both of them inhibit the transport from the subtropics. BAR-15 suppresses horizontal transport from the subtropics into the tropics, and BAR-35 suppresses transport from the subtropics to the extratropics. Air 35  parcels entering the barrier along their trajectories during one model time step are moved to their starting locations after

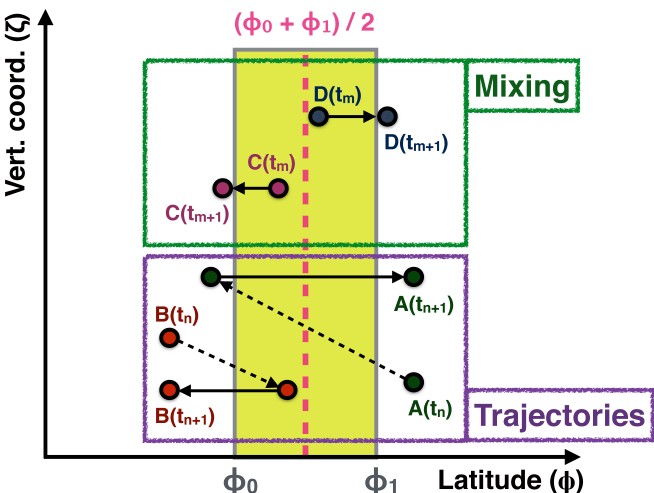

**Figure 1.** A schematic of the implementation of transport barriers in the CLaMS model into the trajectory and mixing modules respectively. The x-axis represents latitude, the y-axis is the vertical coordinate, respectively. The barrier is shown in light-green colour, between $\phi_0$ and $\phi_1$ latitudes. The capital letters A, B, C and D represent the cross-barrier movements of different air parcels between time steps $t_n$ and $t_{n+1}$ for the trajectory module, and $t_m$ and $t_{m+1}$ for the mixing module.

$\Delta t$, as shown in Figure 1. Air parcels which were mixed into the barrier after the mixing procedure are moved to the closer barrier edge after the time step. Because of the broad barrier width of $10°$, this technique inhibits all cross-barrier transport. The CLaMS mixing parametrization ensures that no unrealistic clustering of air parcels occurs at the barrier edges. Third, to investigate the effects of small-scale mixing, we vary the parametrized mixing strength in CLaMS. A discussion of the choice of the critical Lyapunov coefficient, controlling the strength of small-scale mixing in CLaMS, is given by Riese et al. (2012); Konopka et al. (2005b). Hence, for a horizontal resolution of $100\,km$ and a mixing step of 24 h, which were used also in our study, Lyapunov coefficients of $1.5\,day^{-1}$ and $1.2\,day^{-1}$ provide good agreement between observations and simulation results, as found from comparison of CLaMS simulations with observations from infrared limb-sounding from the research aircraft Geophysica (Khosrawi et al., 2005). In particular, using the value of $1.2\,day^{-1}$ gives a better agreement with observations in the 2D version of CLaMS (Konopka et al., 2003). Furthermore, Konopka et al. (2004, 2005b) showed that the value of $\lambda_c = 1.5\,day^{-1}$ (corresponding to the critical deformation of $\gamma_c = 1.5$) for the chosen horizontal resolution and time step here, turns out to be optimal for the 3D version of CLaMS. Even for such a small difference in the small-scale mixing strengths, annual mean $H_2O$ concentrations in the extratropical LS differs by about 10-15% (Riese et al., 2012; McKenna et al., 2002a).

In our study we use a value of $\lambda_c = 1.5\,day^{-1}$ for the reference run, $2.0\,day^{-1}$ to represent weak mixing, and $1.0\,day^{-1}$ for modelling strong mixing to cover the range of realistic small-scale mixing strength. Furthermore, we carry out a simulation without small-scale mixing (mixing in CLaMS was switched off), equivalent to a critical Lyapunov exponent of infinity. The

large range of chosen mixing parameters here ($\lambda_c$) enables investigating sensitivities throughout a large range of possible mixing strengths, including significantly changed mixing characteristics in a potential future climate. In addition, we estimated the vertical diffusivity coefficient for the TTL for the different model simulations. The result suggests a non-linear response of $H_2O$ to the small-scale mixing in CLaMS (details are considered in Sect. 4).

Note again, that small-scale mixing in CLaMS is parametrized in a physical way, by coupling the mixing intensity to deformations in the large-scale flow. The sensitivity of simulated $H_2O$ to the parametrized mixing strength can therefore be regarded representative to the response of changes in small-scale turbulence, as well as to the response of changes in numerical diffusion in climate models.

## 2.2   Satellite observations

We use satellite observations from Aura Microwave Limb Sounder (MLS) and Atmospheric Chemistry Experiment-Fourier Transform Spectrometer (ACE-FTS) for validation of the CLaMS simulations. For MLS, we use Level 2 data of Version 4. Detailed information on the MLS instrument can be found in Waters et al. (2004), and a general discussion of the microwave sounding technique is given in Waters et al. (1999). The MLS instrument was launched on 15 July 2004 on the NASA Aura satellite and measured limb emissions in broad spectral regions. Vertical profiles are retrieved every 165 km along the suborbital

track, covering $82°$ S to $82°$ N latitudes on each orbit. Generally, MLS measurements include around 15 atmospheric chemical species along with temperature geopotential height, relative humidity (deduced from the $H_2O$ and temperature data), cloud ice water content and cloud ice water path, all described as functions of pressure. All measurements are made simultaneously and continuously, during both day and night (Waters et al., 2006). The resolution of the retrieved data is strictly related to the averaging kernels (Rodgers, 2000), which describe both vertical and horizontal resolution. Particularly, the vertical resolution

for $H_2O$ is around 3 km in the UTLS region, whereas the along-track horizontal resolution is in between 170 and 350 km (Livesey et al., 2017).

    As a second satellite observation we used ACE-FTS Version 3.6. ACE-FTS is a part of a Canadian satellite mission for remote sensing of the Earth's atmosphere, SCISAT, which was launched into low Earth circular orbit on 12th of August 2003. ACE-FTS is a satellite instrument which covers the spectral region from 750 to $4400\,cm^{-1}$, and works mainly in solar

occultation. During sunrise and sunset, the ACE-FTS instrument measures sequences of atmospheric absorption spectra in the limb viewing geometry. Further, the spectra are analysed and inverted into vertical profiles. Aerosols and clouds are being monitored using the extinction of solar radiation. The satellite provides altitude profile information (typically from 10 to 100 km) for temperature, pressure, and the volume mixing ratios for several molecule species over the latitudes from $85°$ N to $85°$ S (Bernath et al., 2005). Solar occultation instruments like the ACE-FTS could have a high vertical resolution as good as $\approx$

1 km, but low horizontal resolution ($\approx$ 300 km) in the limb direction (Hegglin et al., 2008). A detailed description of ACE-FTS is given by Bernath (2017).

    Hurst et al. (2016) compare MLS $H_2O$ observations in the LS with balloon-borne Cryogenic Frostpoint Hygrometer (CFH) and Frost Point Hygrometer (FPH) instruments, from 2004 to 2015. There is a potential drift between the two sets of measurements, with MLS $H_2O$ increasing at a rate of around 0.03-0.07 ppmv $yr^{-1}$ relative to the hygrometer measurements, starting

around 2009. In contrast, the comparisons with recent ACE-FTS data show no signs of such drift in MLS $H_2O$, nor do comparisons of MLS upper stratospheric $H_2O$ with ground-based microwave measurements (Livesey et al., 2017).

## 3 Results

Figure 2 shows the annual cycle of tropical ($10°S-10°N$) stratospheric entry $H_2O$ at $400\,K$ for all simulations. While a clear
annual cycle is evident for all cases, the mixing ratios vary by more than $1\,ppmv$ between the simulations. The reference simulation (REF) agrees well with MLS data, although there are some small differences during boreal winter. The largest sensitivity (spread between simulations) occurs for boreal summer and fall months. Suppressing horizontal transport from the subtropics into the tropics (BAR-15) significantly dries the tropical entry $H_2O$, with difference to the reference of up to around $1\,ppmv$. For the mixing sensitivity simulations, the largest difference from the reference case occurs for the case without mixing
(MIX-no), with the MIX-no simulation dryer by about $\approx 0.8\,ppmv$ in September-October. The CLaMS simulation driven with JRA-55 shows moister values in the TTL compared to the ERA-Interim simulation, in agreement with the recent findings of Davis et al. (2017) (for details see Sect. 4).

The strong sensitivity of tropical entry $H_2O$ shows that the TTL temperatures from the used reanalysis dataset, horizontal transport and small-scale mixing are critical control factors for stratospheric $H_2O$. They will be investigated in more detail in
the following.

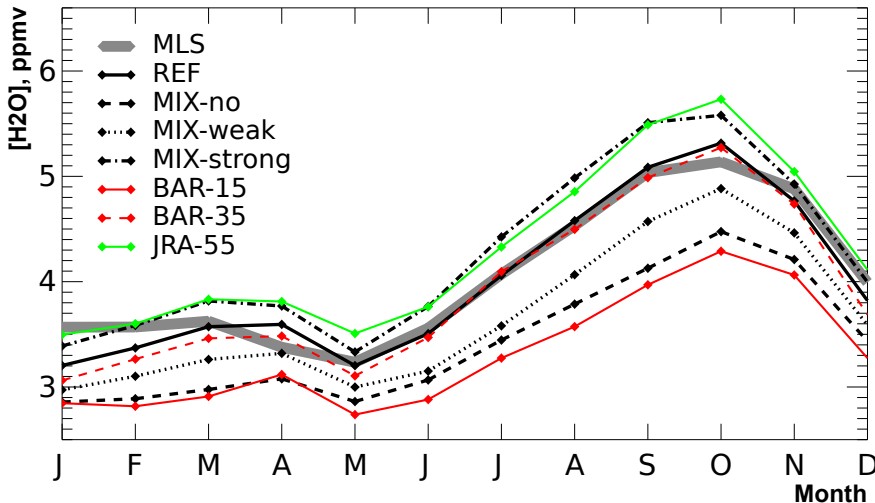

**Figure 2.** Annual cycle of tropical entry $H_2O$ at $400\,K$ ($10°S-10°N$) from different sensitivity simulations with respect to variations in reanalysis datasets, horizontal transport and small-scale mixing for 2011. Grey line represents MLS satellite observations, for comparison. Shown are the reference simulation (REF), the cases without mixing (MIX-no), with weak (MIX-weak) and strong mixing (MIX-strong), the simulations with transport barriers at $15°N/S$ (BAR-15), at $35°N/S$ (BAR-35), and the simulation driven with JRA-55 reanalysis data (JRA-55).

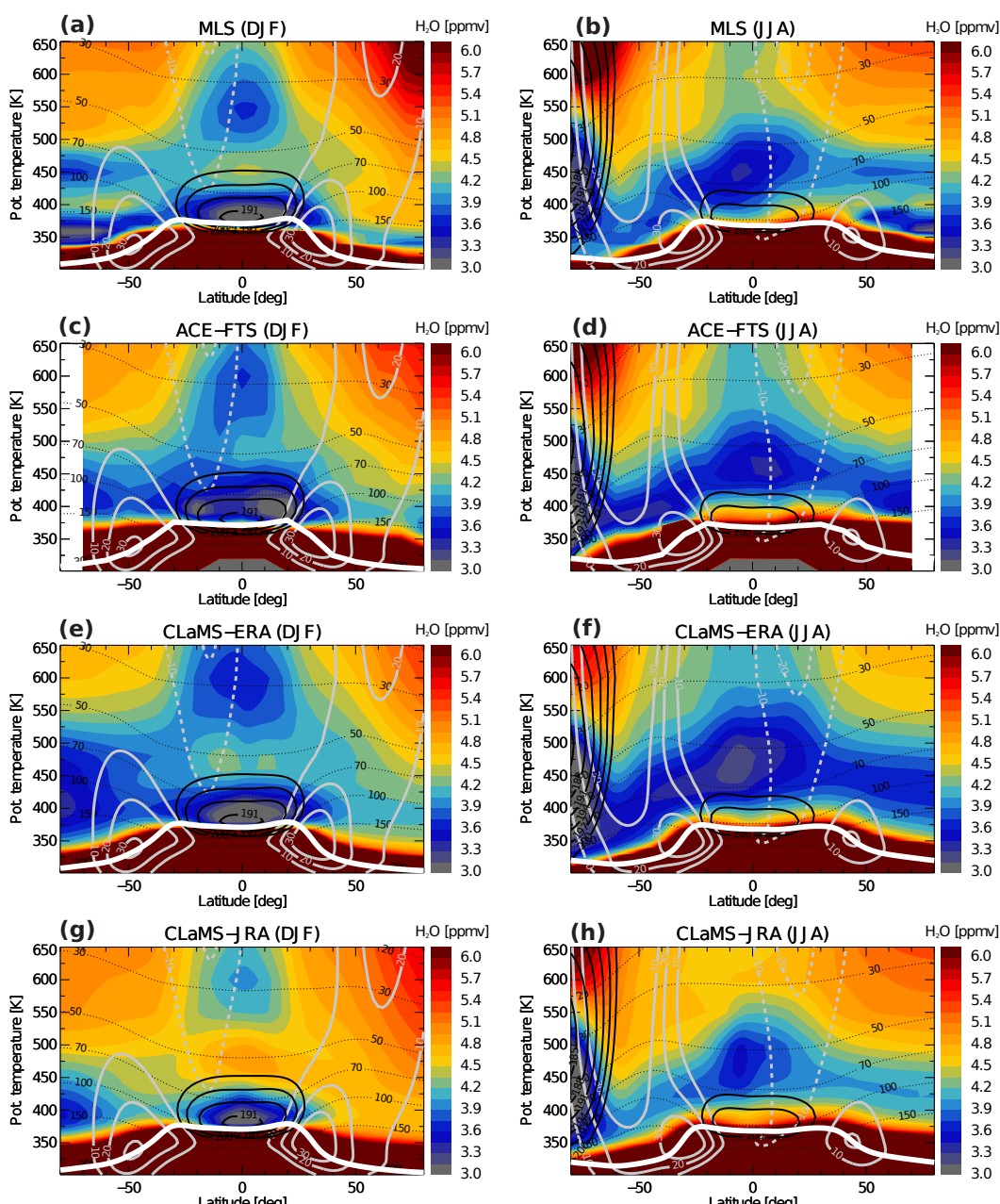

**Figure 3.** Zonal mean H$_2$O distributions for winter (DJF, left) and summer (JJA, right) from MLS and ACE-FTS satellite observations, as well as for CLaMS simulations driven with either ERA-Interim or JRA-55 reanalysis. Data shown are climatologies for 2004-2013 years. Black contours show temperatures (185 K, 188 K, 191 K, 194 K, 197 K, 200 K), grey contours are zonal winds (10 m/sec, 20 m/sec, 30 m/sec), black dotted lines are pressure levels (in hPa) and the white line is the thermal tropopause.

## 3.1 Reanalysis uncertainty

Zonal mean $H_2O$ mixing ratios for boreal winter (December-February, DJF) and summer (June-August, JJA) from MLS and ACE-FTS satellite observations, and from CLaMS simulations driven by ERA-Interim and JRA-55 are shown in Figure 3. The comparison of the two different satellite datasets (first and second row) shows differences of about 0.5 ppmv (with ACE-FTS being moister), and even larger in the extratropical LS. Oscillations in MLS $H_2O$ at high latitudes are a known effect of the broad averaging kernel (Ploeger et al., 2013). At low latitudes the effects of the MLS averaging kernel on $H_2O$ are much smaller and we do not apply it to the model data here, in order not to smear out the structure in the simulated $H_2O$.

Comparison between the two simulations, driven by either ERA-Interim or JRA-55 (third and fourth row), shows differences due to the used reanalysis dataset of about 0.5 ppmv, increasing towards the extratropical lowermost stratosphere. The main reason for JRA-55 causing a moister stratosphere, when compared to ERA-Interim is the positive difference in the temperatures around the TTL (Fig. 4). Zonal mean temperatures in this region are on average about 2 K higher for JRA-55 than for ERA-Interim. Remarkably, these differences only exist in a narrow layer around the tropical tropopause. In addition to temperature, also differences in winds and heating rates between the two reanalysis could cause differences in $H_2O$ mixing ratios, however, the temperature difference provides a self-evident explanation.

A detailed comparison of the LS $H_2O$ between MLS and CLaMS simulations driven by ERA-Interim and JRA-55 reanalysis at 380 K is given in Figure 5. Note, that the 380 K surface may be located well below the tropopause in some regions (e.g., Asian monsoon). The patterns of dominant freeze-drying regions above the West Pacific and South America in boreal winter are consistent between the observations and the two simulations. Notably, the larger area of low $H_2O$ mixing ratios and colder temperatures for ERA-Interim when compared to JRA-55, is consistent with the drier global stratosphere, as discussed above. Also in boreal summer, the $H_2O$ distributions for MLS observations and CLaMS, driven by the two reanalysis, are similar

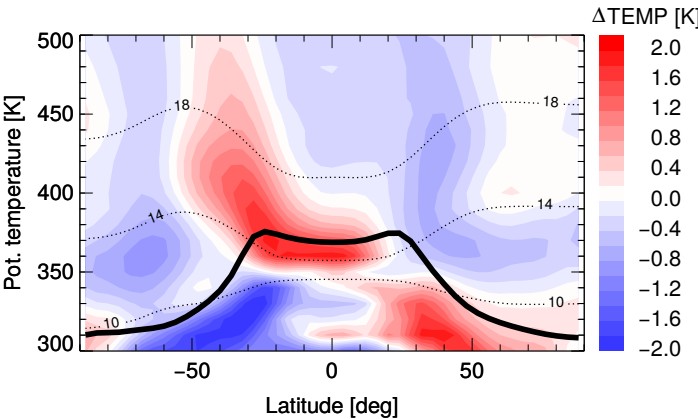

**Figure 4.** Differences in the zonal mean temperatures between JRA-55 and ERA-Interim reanalysis data averaged for the period of 1979-2013; black dotted lines are altitude levels (in km) and the black line is the thermal tropopause.

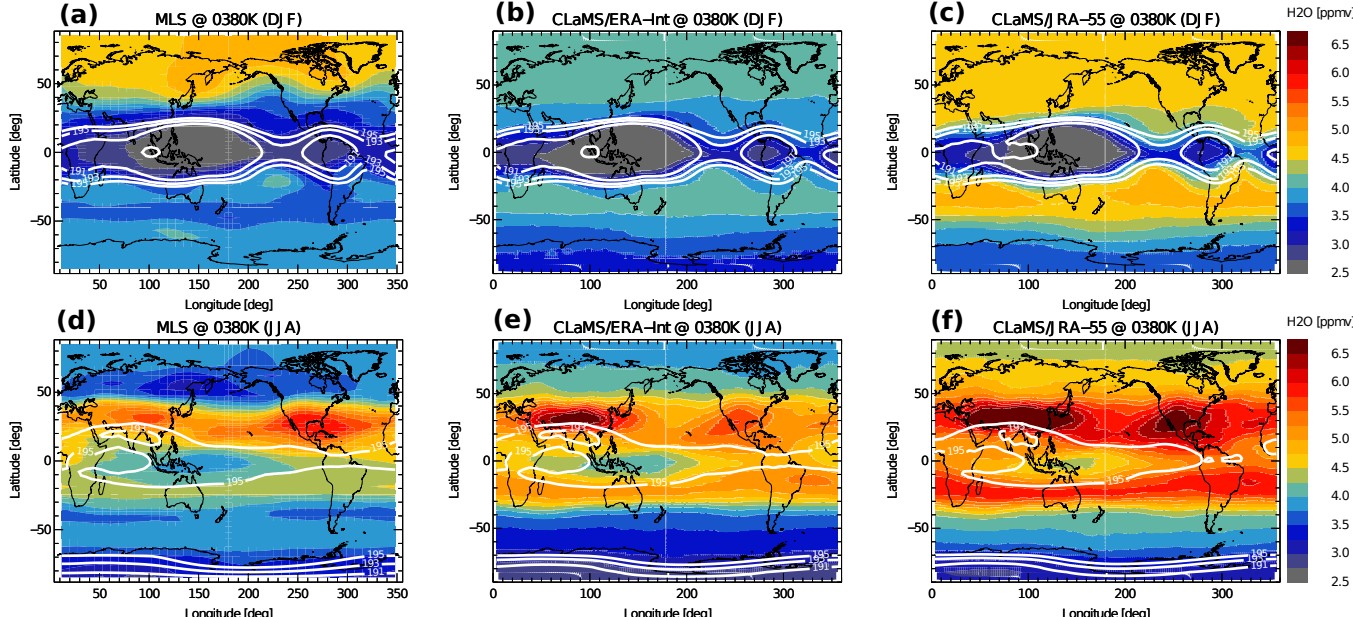

**Figure 5.** Water vapour maps at 380 K from MLS (a, d), and CLaMS simulations driven by ERA-Interim (b, e) and JRA-55 (c, f). Shown are winter (December-February, DJF) and summer (June-August, JJA) data, respectively, from a 2004-2013 climatology. White lines show temperature contours (191 K, 193 K, 195 K). Note, that the temperatures at the (a,b) are from ERA-Interim reanalysis sampled at the MLS locations.

in the tropics. Nevertheless, in the subtropics, the strength of summertime monsoon anomalies in MLS differs from CLaMS, with the Asian monsoon dominating in both simulations, while the American monsoon appears stronger in MLS data (e.g., Ploeger et al., 2013). Note, that the long-term $H_2O$ time series from CLaMS driven by ERA-Interim reanalysis agrees well with the Halogen Occultation Experiment (HALOE) and MLS observations (Tao et al., 2015).

5    Overall, regarding the global $H_2O$ distributions and maps in the LS, CLaMS modelling results with ERA-Interim are drier when compared to JRA-55, resulting from lower TTL temperatures in ERA-Interim. The agreement between CLaMS based on ERA-Interim and JRA-55 with the observations strongly depends on the considered region and season. And it is not possible to conclude from our analysis which reanalysis results in simulated $H_2O$ in the best agreement with the observations.

### 3.2 Horizontal transport effects

10   Probability density functions (PDFs) of $H_2O$ mixing ratio (e.g., Schoeberl et al., 2013), allow a simple comparison of the overall effects of horizontal transport on $H_2O$ (in the LS) by contrasting the various sensitivity simulations. Figure 6 shows these PDFs for the tropics and extratropics of both hemispheres for the different barrier simulations.

In the SH (Fig. 6a) the frequent very low mixing ratios are insensitive to horizontal transport, indicating the occurrence of local dehydration. This insensitivity reflects the fact that temperatures in the Antarctic polar vortex are so low that $H_2O$ mixing

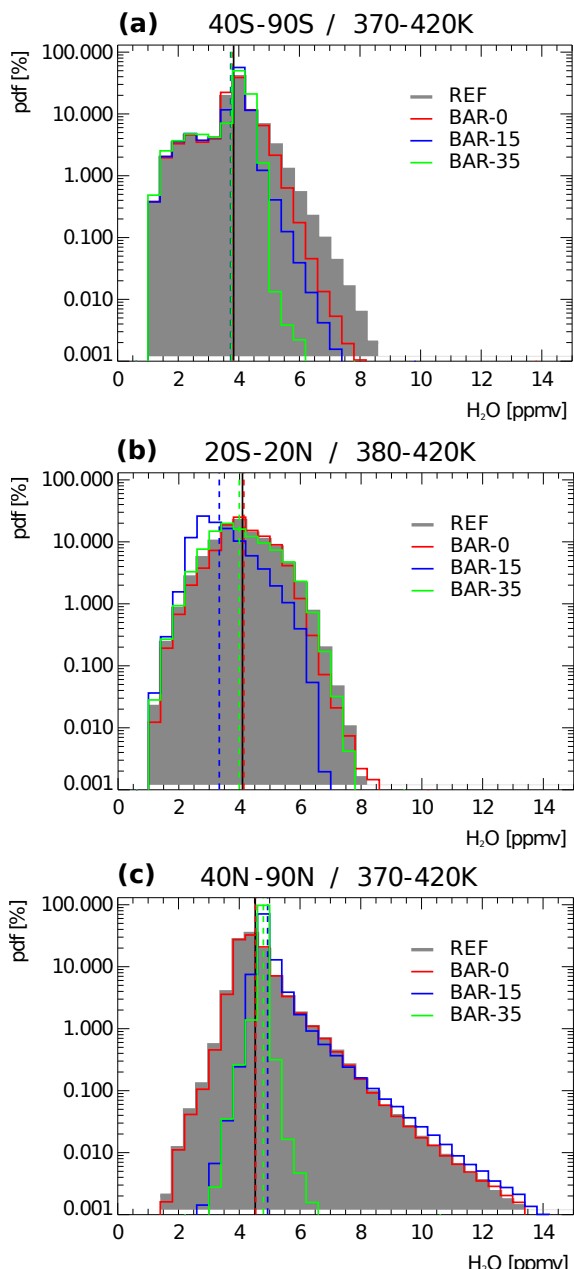

**Figure 6.** Probability density function (PDF) for water vapour mixing ratios in the SH extratropics for 40°S-90°S between 370 K and 420 K (a), in the tropics for 20°S-20°N between 380 K and 420 K (b), and in the NH extratropics for 40°N-90°N between 370 K and 420 K (c). Shown data are from 2011 CLaMS sensitivity simulations with horizontal barriers along latitude circles at 0° (BAR-0, red solid line), 15° N/S (BAR-15, blue solid line), 35° N/S (BAR-35, green solid line) and the reference (REF, grey background). Dashed coloured lines represent the mean $H_2O$ values for the different simulations respectively, whereas the black solid line shows the mean value of the reference simulation.

ratios are locally freeze-dried to the saturation value. However, there is a weak effect of transport from the NH on moistening the SH, indicated by a lowering of the PDF's tail without cross-equatorial transport. Suppressing transport from the tropics lowers the tail further, and suppressing transport from the SH subtropics (with the 35°S barrier) finally removes almost all mixing ratios higher than 5 ppmv.

In the tropics (Fig. 6b), the insignificant difference between the reference (REF) and an equatorial transport barrier (BAR-0) simulations shows that the interhemispheric transport is rather unimportant for tropical mean $H_2O$ mixing ratios. Similarly, in-mixing of mid- and high-latitude air (see BAR-35) has a very small impact on tropical mean $H_2O$, in agreement with the findings of Ploeger et al. (2012). In contrast, transport from the subtropics into the tropics has a strong effect. Suppressing such transport by applying a barrier at 15° N/S (BAR-15) changes the PDF substantially, as evident from the difference between the

simulation BAR-15 and the reference cases. The isolation of the tropics due to the lack of horizontal transport in the BAR-15 simulation (all the way from the surface to 600 K) between equator and subtropics (both ways) causes dry air at the equator. Thus, with the barrier at 15° N/S the fraction of dry air at the equatorial region increases. The comparison of BAR-15 with BAR-35 shows that transport from the subtropical region into the tropics increases $H_2O$. Without transport from the subtropics, also the tropical mean $H_2O$ PDF appears more strongly skewed towards low mixing ratios (blue line in Fig. 6b), and the mean

$H_2O$ mixing ratio is shifted towards lower values by about 0.5 ppmv.

In the NH cross-equatorial transport from the SH has only a weak effect, as visible from the equatorial transport barrier (Fig. 6c). Introducing a transport barrier in the subtropics at 15°N removes the low mixing ratios from the PDF, showing that these low mixing ratios result from transport out of the deep tropics. Moving the transport barrier further away from the equator to 35°N, changes the PDF drastically. In addition to the low mixing ratios it also removes the tail of the PDF at high mixing

ratios, such that a very narrow extratropical $H_2O$ mixing ratio PDF remains. Hence, these high mixing ratios are the result of transport from the subtropics, and are likely related to the Asian monsoon, as argued by Ploeger et al. (2013). Accordingly, monsoon driven $H_2O$ transport from the subtropics to the high latitudes is by far more pronounced for the NH than for SH.

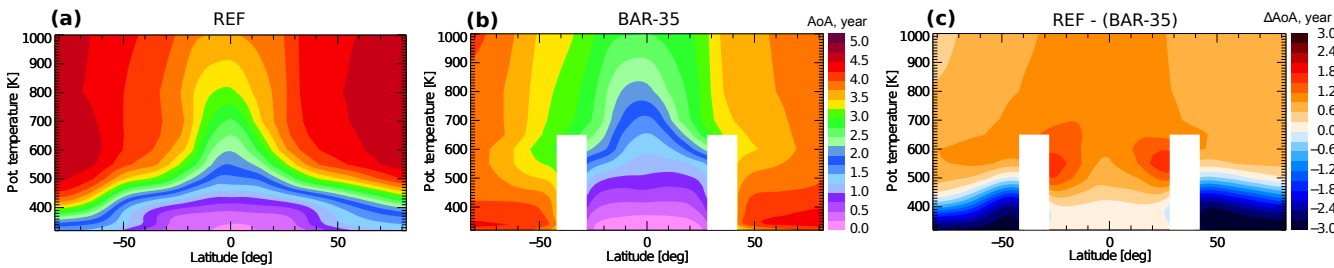

**Figure 7.** Zonal mean age of air distributions for 2011. Shown data are from the CLaMS reference simulation (a) and from the sensitivity simulation with barrier along latitude circles at 35° N/S (b) and the absolute difference between them (c). Transport barriers are set between the Earth's surface and the 600 K potential temperature levels and are represented in white.

The pure transport effects of horizontal exchange between tropics and mid-latitudes are evident from mean age of air (AoA), the mean transit time for air through the stratosphere for the different model experiments with horizontal transport barriers. Figure 7 shows CLaMS calculations of the AoA for the reference case (Fig. 7a), simulation with transport barriers in the subtropics at 35° N/S (Fig. 7b) and the absolute difference between them (Fig. 7c). These horizontal transport barriers at 35° N/ effectively isolate the tropical pipe from the in-mixing of older stratospheric air from mid-latitudes significantly decreasing the AoA globally by more than a year. Hence, recirculation from mid-latitudes into the tropics has a strong ageing effect on the stratosphere globally, in agreement with (Neu and Plumb, 1999). Without recirculation (in the BAR-35 simulation) the global AoA distribution reflects mainly the pure effect of the residual circulation, resulting in oldest air in the extratropical lowermost stratosphere, and appears very similar to the distribution of residual circulation transit times (e.g., Ploeger et al., 2015). Older air in the NH is related to the deeper NH residual circulation cell.

In the tropics, the age distribution in Fig. 7b shows a weak double peak structure up to about 500 K, indicating that the subtropics are regions of particularly fast transport, likely related to subtropical processes like monsoon circulations. Suppressing transport in the subtropics with barriers at 35° N/S therefore significantly increases AoA in the extratropical stratosphere (Fig. 7c). A similar result has been recently found by Garny et al. (2014). Furthermore, Garny et al. (2014) presents a nice explanation of the recirculation process, describing recirculation as a process when an air parcel enters the tropical stratosphere and travels along the residual circulation to the extratropics, where it can be mixed back into the tropics, and thus recirculates along the residual circulation again. In this way, the age of air of the parcels increases steadily while performing multiple circuits through the stratosphere.

Relating the pure horizontal transport effects seen in age of air to $H_2O$ is not straightforward, as $H_2O$ is strongly controlled by TTL temperatures. Figure 8 shows the annual zonal mean $H_2O$ mixing ratio for the different sensitivity simulations with transport barriers, and Figure 9 highlights the differences between the simulations with largest $H_2O$ changes. The small differences between the reference (Fig. 8a) and the equatorial barrier (Fig. 8b) simulations indicate only a very weak effect of transport processes in the deep tropics and interhemispheric exchange on global stratospheric $H_2O$. Similarly, the sensitivity simulation with a subtropical transport barrier at 35° N/S (Fig. 8c, 9c) shows that in-mixing of mid-latitude air has only a weak impact on global stratospheric $H_2O$ (except in the NH LS). In contrast, transport from the subtropics, between 10° and 30° N/S, as visible from the comparison of sensitivity simulation BAR-15 and BAR-35 (Fig. 2, 8c, 8d, 9a), has a strong effect on tropical entry $H_2O$ and hence on global $H_2O$. Without such transport from the subtropics (Fig. 8a, 8d), the stratosphere becomes substantially drier (maximal differences in the stratosphere are up to about 1 ppmv). The fact that this drying occurs only with transport barriers at 15° N/S and not with barriers at 35° N/S, shows that it is not related to the suppression of recirculation of aged air from mid-latitudes, which has been affected by methane oxidation. In fact, processes in the subtropics (e.g., monsoon circulations) have a strong effect in moistening the global stratosphere, and suppressing these processes in BAR-15 causes drying. The model experiments with transport barriers only in the NH or SH subtropics further show that the effect of the NH subtropics in moistening the global stratosphere is stronger compared to the SH subtropics (Fig. 9b).

Figure 10 shows the $H_2O$ seasonal cycle at 400 K and its latitudinal structure, sometimes termed the "horizontal tape-recorder" (e.g., Randel et al., 2001; Flury et al., 2013). Consistent with the discussion above, a transport barrier at the equator

has only a very weak drying effect on the SH subtropics and mid-latitudes, indicating only a minor role of the NH in moistening the SH lowest stratosphere. Furthermore, the effect of horizontal transport on the SH is small in all simulations, as H$_2$O mixing ratios in the SH are strongly affected by local freeze-drying at SH high-latitudes. In the NH, horizontal transport moistens the extratropical LS in summer, and dries this region in winter. In the tropics, the annual cycle is related to minimum tropopause temperatures during boreal winter and maximum tropopause temperatures during summer. Therefore, during winter, horizontal

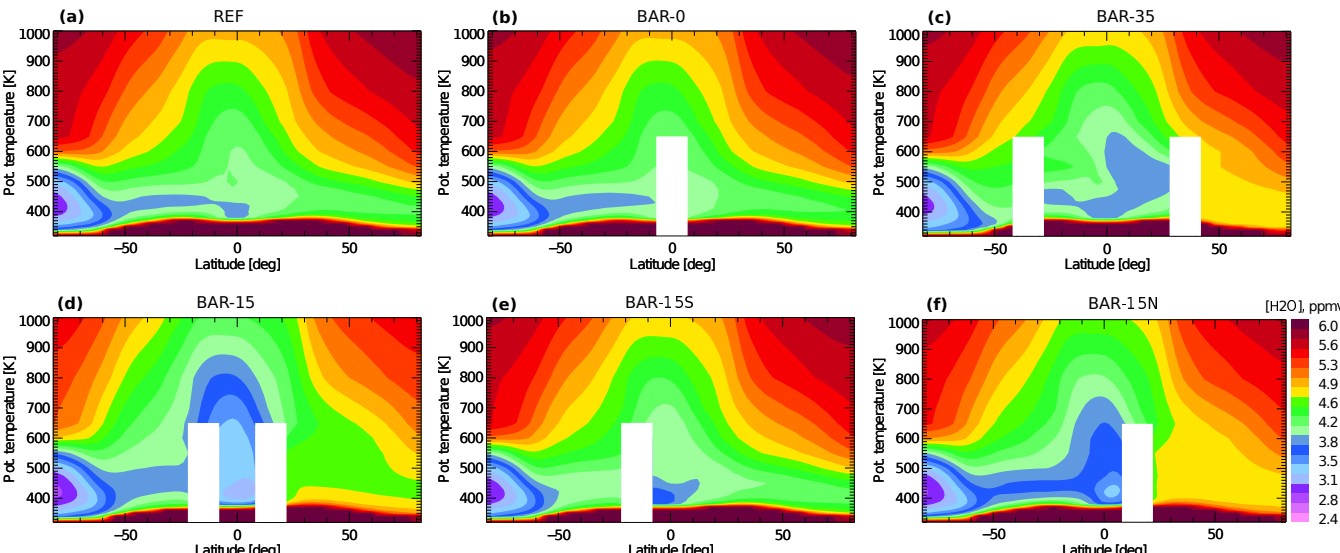

**Figure 8.** Zonal mean water vapour distributions for 2011. Shown data are from the CLaMS sensitivity simulations for the reference (a) and the horizontal transport barrier simulations along latitude circles at 0° (b), 35° N/S (c), 15° N/S (d), 15° S (e) and 15° N (f). Transport barriers are set between the Earth's surface and the 600 K potential temperature levels and are represented in white.

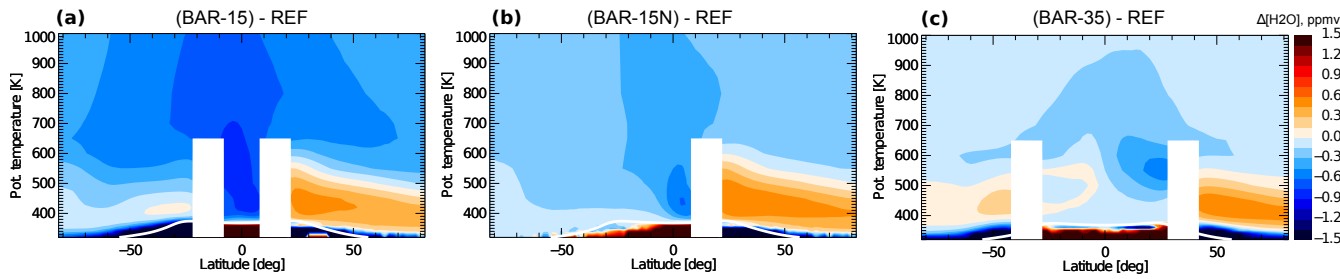

**Figure 9.** Zonal mean water vapour distributions for 2011. Shown are differences between the CLaMS reference simulation and the sensitivity simulations with the horizontal transport barriers at 15° N/S (a), 15° N (b), and 35° N/S (c). Transport barriers are set between the Earth's surface and the 600 K potential temperature levels and are represented in white. Tropopause is presented as white solid line, and is calculated from ERA-Interim reanalysis data.

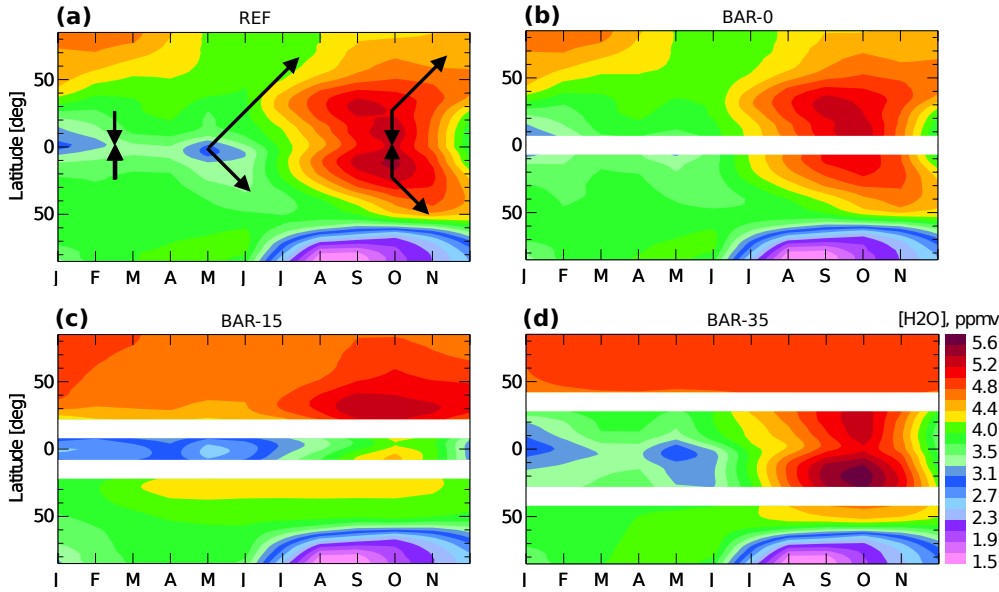

**Figure 10.** Horizontal tape-recorder of water vapour at the potential temperature level θ = 400 K for 2011. Shown data are from CLaMS sensitivity simulations for the reference (a) and the sensitivity simulations with transport barriers along latitude circles at 0° (b), 15° N/S (c) and 35° N/S (d). Transport barriers are represented in white, the main directions of air mass transport are presented with black arrows.

transport exports dry air out of the tropics into the NH, and, during summer, moist air. Consequently, the entire annual cycle of the $H_2O$ in the NH extratropical LS is related to horizontal transport from low latitudes, as argued by Ploeger et al. (2013). The boreal summer maxima are related to monsoonal circulations and transport out of the tropics, along the eastern and western flanks (Randel and Jensen, 2013).

## 3.3 Mixing effects

The PDF of $H_2O$ mixing ratio in Figure 11 shows that increased small-scale mixing in the model generally moistens the LS in the tropics, as well as in the extratropics of both hemispheres. Increased mixing causes both a decrease in the fraction of dry air and an increase in the fraction of moist air, and therefore shifts the PDF to higher mixing ratios. In particular, for the NH extratropics, this effect is strong, substantially enhancing the tail of the PDF, with simultaneously reducing the low values in the PDF. The mean $H_2O$ mixing ratio is also increasing towards higher values with increasing mixing strength (dashed lines in Fig. 11).

Changes in the parametrised small-scale mixing strength, however, may affect different processes, critical to the distribution of $H_2O$ in the LS region. Such processes are: diffusive cross-tropopause moisture transport, recirculation of air masses, permeability of the tropical pipe, and vertical diffusion (for illustration see Fig. 12). Therefore, interpreting the mixing effects in terms of processes is a challenging task.

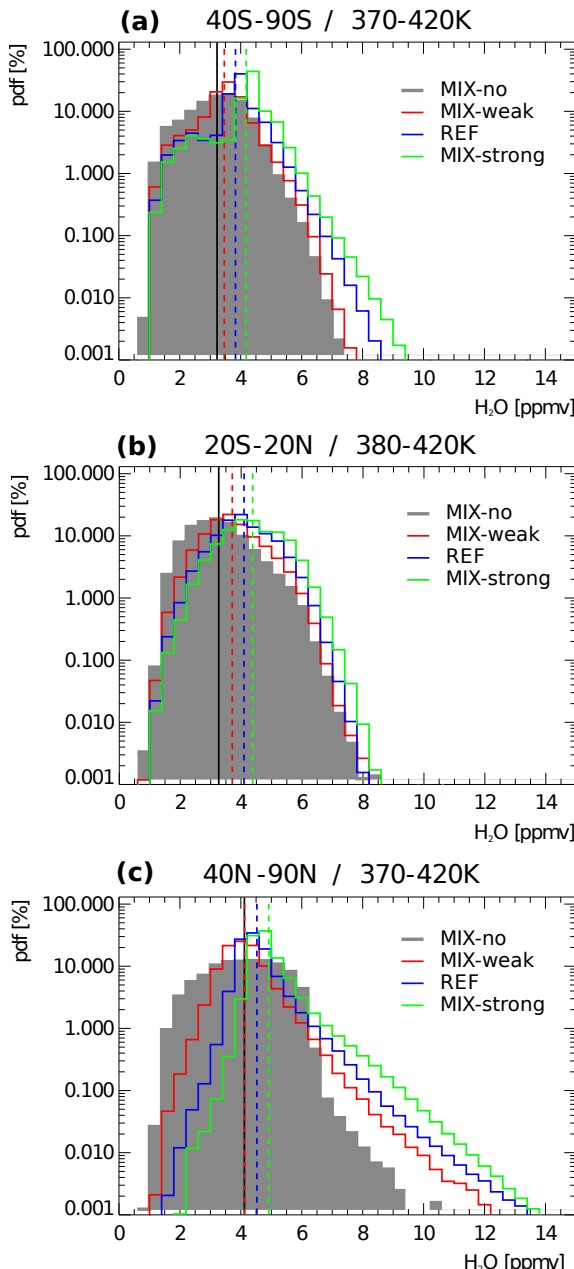

**Figure 11.** Probability density function (PDF) for water vapour mixing ratios in the SH extratropics for 40°S-90°S between 370 K and 420 K (a), in the tropics for 20°S-20°N between 380 K and 420 K (b), and in the NH extratropics for 40°N-90°N between 370 K and 420 K (c). Shown data are from 2011 CLaMS sensitivity simulations with different strength of small-scale mixing for the case without mixing (MIX-no, grey shading background), weak mixing (MIX-weak, red solid line), reference simulation (REF, blue solid line) and strong mixing (MIX-strong, green solid line). Dashed coloured lines represent the mean $H_2O$ values for the different simulations respectively, whereas the black solid line shows the mean value for the non-mixing case.

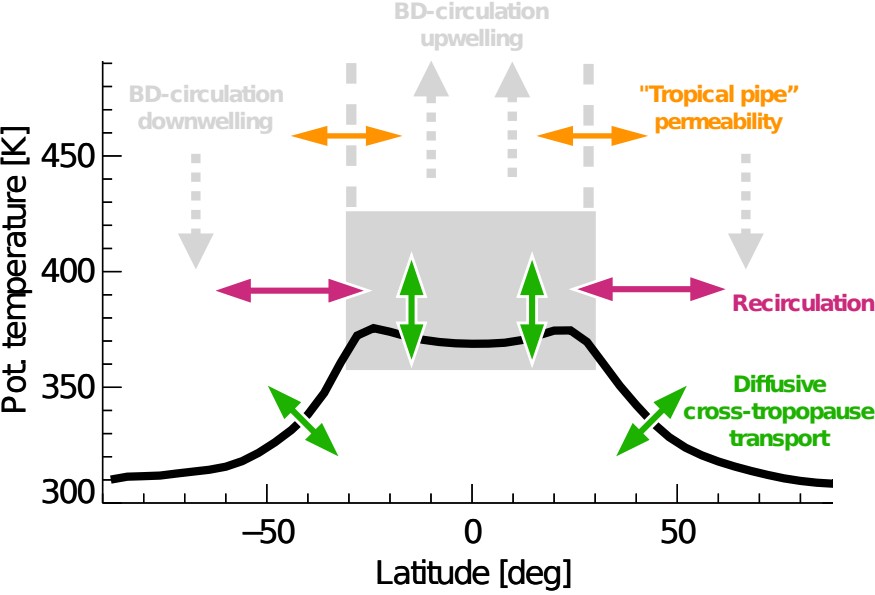

**Figure 12.** A schematic of considered processes critical for the distribution of water vapour in the LS region. Grey arrows represent Brewer-Dobson upwelling throughout the tropical region and downwelling towards the poles; green arrows stand for cross-tropopause transport through the whole latitudinal range; the pink arrows represent the region of recirculation of air masses in TTL and the orange arrows show the regions of "tropical pipe" permeability. TTL is represented with solid grey background.

Figure 13 shows annual zonal mean distributions of $H_2O$ (a-d) and total water (e-h). Similarly, Figure 14 shows double methane mixing ratios (a-d) and AoA (e-h) from the CLaMS simulation without mixing (MIX-no) and their incremental differences between the sensitivity simulations with increasing mixing from the weak mixing case (MIX-weak) through the reference (REF) to the strong mixing case (MIX-strong); i.e. [(MIX-weak) - MIX-no], [REF - (MIX-weak)], and [(MIX-5  strong) - REF]. Note, that the simulation without small-scale mixing (MIX-no) should not be considered as a realistic case, as turbulent mixing processes always take place in the atmosphere. However, we show the results from this simulation for the sake of completeness, when analysing the mixing effects, and for facilitating comparisons with pure trajectory studies (e.g., Fueglistaler and Haynes, 2005; Schoeberl and Dessler, 2011).

A clear response to mixing is found for the LS (below $\approx 430\,\mathrm{K}$), which is moistened with increasing small-scale mixing. In10  the following, we consider total water above the tropical tropopause as an indicator of changes in transport because it is not affected by chemistry (here methane oxidation). As the moistening in the LS below $430\,\mathrm{K}$ is also evident in total water, but not in methane and mean age, it is largely related to enhanced diffusive cross-tropopause transport of moist air. This enhanced diffusive cross-tropopause transport in turn increases the probability to by-pass the regions of cold temperatures, rendering the freeze-drying at the tropical tropopause less efficient. Consequently, $H_2O$ entering the stratosphere is enhanced with increased

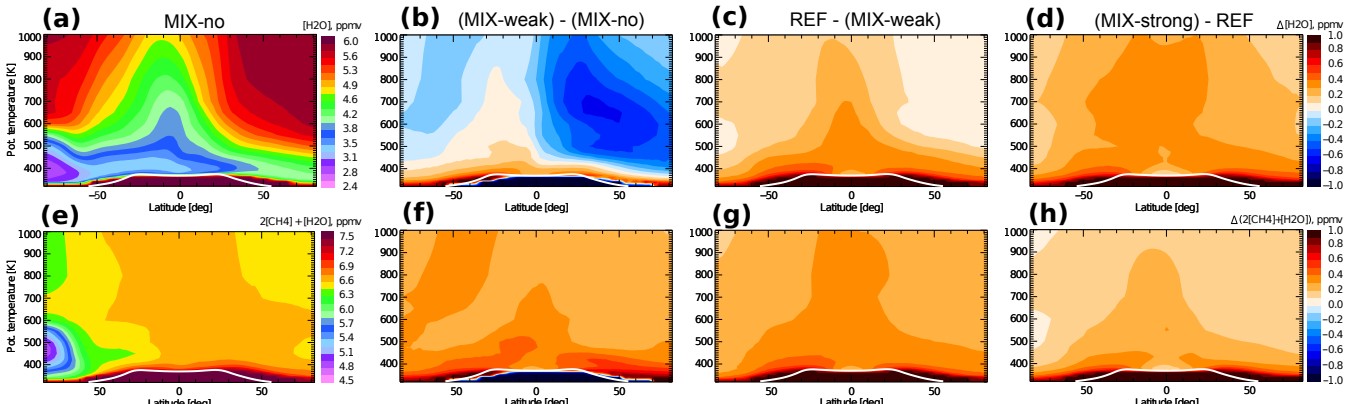

**Figure 13.** Annual zonal mean distributions of water vapour (a) and total water (e) from the CLaMS simulation without mixing (MIX-no), as well as the incremental differences between the sensitivity simulations with increasing mixing from the weak mixing case (MIX-weak) through the reference (REF) to the strong mixing case (MIX-strong); i.e. [(MIX-weak) - MIX-no] in the second column (b, f), [REF - (MIX-weak)] in the third column (c, g), and [(MIX-strong) - REF] in the fourth column (d, h). Tropopause is presented as white solid line, and is calculated from ERA-Interim reanalysis data. The data are shown for 2011.

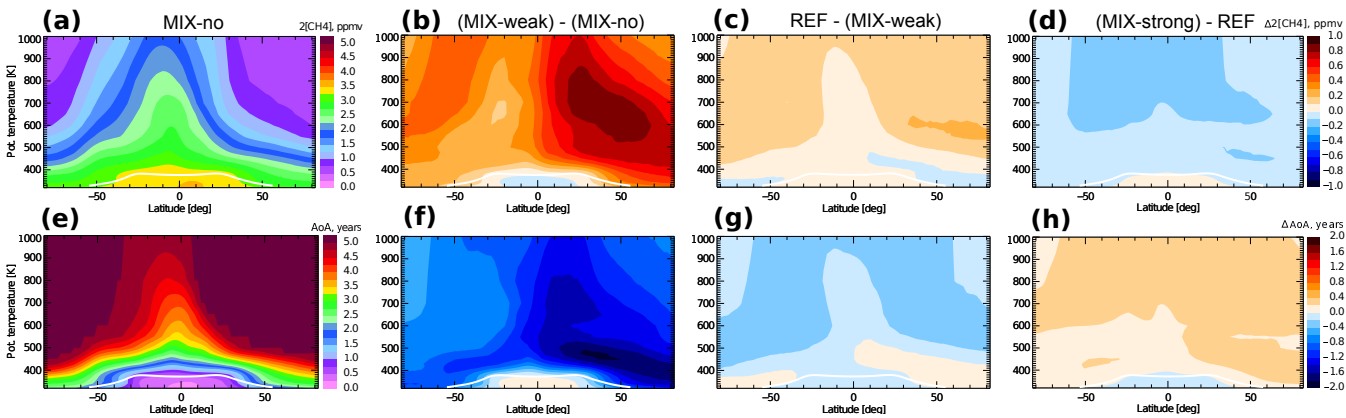

**Figure 14.** Annual zonal mean distributions of $2CH_4$ (a) and mean age of air (b) from the CLaMS simulation without mixing (MIX-no), as well as the incremental differences between the sensitivity simulations with increasing mixing from the weak mixing case (MIX-weak) through the reference (REF) to the strong mixing case (MIX-strong); i.e. [(MIX-weak) - MIX-no] in the second column (b, f), [REF - (MIX-weak)] in the third column (c, g), and [(MIX-strong) - REF] in the fourth column (d, h). Tropopause is presented as white solid line, and is calculated from ERA-Interim reanalysis data. The data are shown for 2011.

small-scale mixing. The response of total water to changes in mixing is largely independent on the reference mixing strength

throughout the stratosphere, with total water always increasing with increased mixing (Fig. 13f, g, h), reflecting the fact that the efficiency of freeze-drying at the tropical tropopause decreases with increasing mixing.

Above about 430 K, the response of $H_2O$ mixing ratio to varying the mixing strength turns out to be more challenging to interpret, and strongly depends on the reference strength of mixing, due to a complex interplay between horizontal and vertical mixing processes. First, increasing the mixing strength from no-mixing (MIX-no) to weak mixing (MIX-weak) causes significant drying in the NH (Fig. 13b). A related signal (above $\approx 430$ K in the middle and upper stratosphere) is evident in methane (Fig. 14b) and mean age (Fig. 14f), but not in total water (Fig. 13f). Hence, the drying response in the NH is attributable to transport effects, most likely to an increased permeability of the tropical pipe with increasing mixing and related increased transport of dry air and enhanced methane out of the tropics and into the NH. Second, increasing the mixing strength from weak mixing (MIX-weak) to reference mixing (REF) and from reference mixing to strong mixing (MIX-strong) causes a moister stratosphere globally, related to enhanced diffusive cross-tropopause transport, and less efficient freeze-drying (see discussion above). For the former (increasing mixing from MIX-weak), a weak increase of methane in the NH (Fig. 14c) indicates a simultaneous increase in the permeability of the tropical pipe. For the latter (increasing mixing from REF), decreasing methane mixing ratios (Fig. 14d) and increasing mean age (Fig. 14h) throughout the stratosphere, likely indicate a simultaneous increase in the strength of recirculation, due to increasing mixing.

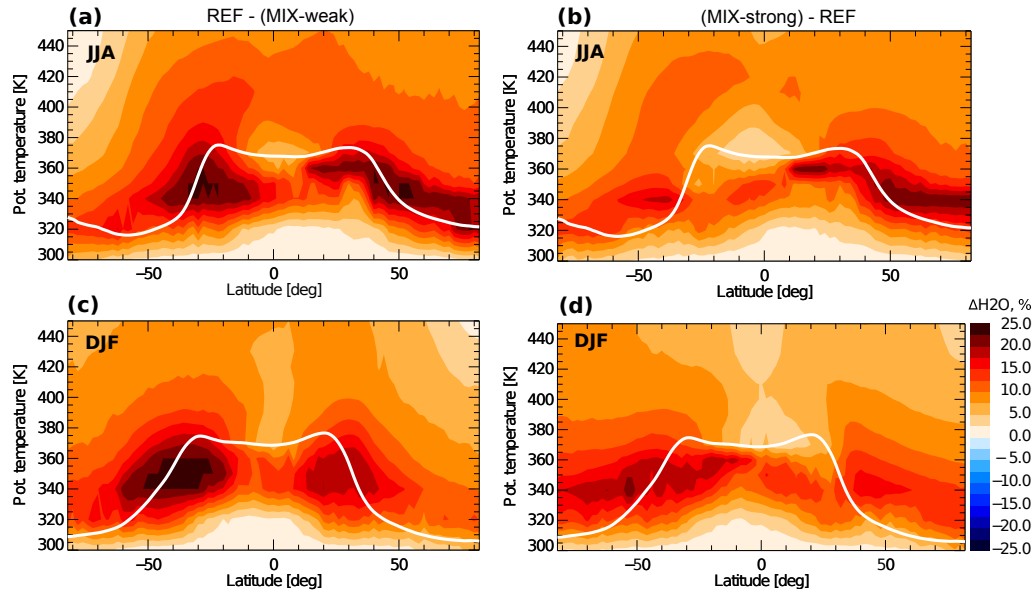

**Figure 15.** Relative differences of zonal mean water vapour for summer (a, b) and winter (c, d) seasons for 2011 between the reference and weak mixing and between the strong mixing and reference case for the LS region. Tropopause is presented with white solid line, and is calculated from ERA-Interim reanalysis data.

Figure 15 presents a zoom-in view onto the $H_2O$ response to mixing changes in the UTLS region, a critical region for global climate, for both summer (a, b) and winter (c, d). Clearly, enhanced small-scale mixing moistens the LS, due to enhanced diffusive cross-tropopause moisture transport, with maximum differences between the simulations of around 20%. The moistening effects are particularly large in the region around the tropopause, where the radiative effect is most sensitive (e.g., Riese et al., 2012). Furthermore, the moistening effect due to mixing maximises in the summer hemisphere (Konopka et al., 2007). In the SH, $H_2O$ in the subtropical jet regions appears to be most critical to changes in small-scale mixing. In particular, increasing $H_2O$ mixing ratios in the extratropical lowermost stratosphere cause a flattening of the $H_2O$ isopleths towards high latitudes.

Maps of the $H_2O$ distribution at 380 K for the different CLaMS simulations with different small-scale mixing strength show the regions most prone to mixing changes (Fig. 16). Strongest moistening, due to increased mixing, occurs in the regions of subtropical jets. This is consistent with the findings of Konopka and Pan (2012), showing that the subtropical jets are regions of intense mixing. Most intense moistening is always caused by mixing processes along the subtropical jet in the summer hemisphere. During boreal winter the SH subtropical jet substantially moistens with increasing mixing, during boreal summer the NH jet moistens. In particular, the moist anomaly of the Asian and American monsoons during boreal summer is affected by small-scale mixing. Without mixing only a weak anomaly occurs in the Asian monsoon, while the moist anomaly in the American monsoon is absent. With increased mixing the Asian monsoon moist anomaly first increases (MIX-weak and REF cases). When mixing becomes very strong (MIX-strong) this behaviour changes and the entire jet region becomes strongly moistened such that the Asian monsoon moist anomaly relative to the entire jet region weakens. It should be noted here that

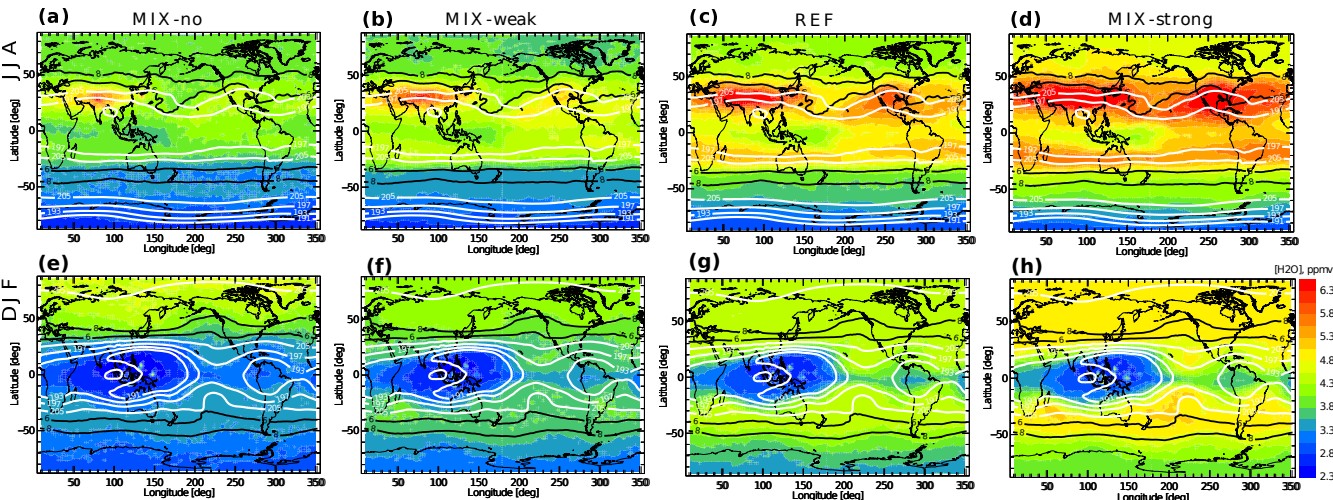

**Figure 16.** Seasonal mean water vapour distribution for 2011 from the CLaMS simulation without mixing (MIX-no) and the sensitivity simulations with non-vanishing mixing strength (MIX-weak, REF, MIX-strong) at the potential temperature level of 380 K; DJF indicates winter (a-d), and JJA summer periods (e-h). Grey lines are potential vorticity from ERA-Interim reanalysis data (6 PVU, 8 PVU), and the violet ones are temperatures (191 K, 193 K, 197 K, 205 K) taken from ERA-Interim reanalysis data

the 380K potential temperature surface may well be located below the tropopause in the Asian monsoon region, such that parts of the moist anomaly in Fig. 16 indicates tropospheric air rather than cross-tropopause transport. However, the response of the subtropical jet and monsoon moist anomalies to increased small-scale mixing remains comparable also at 400K and hence appears to be related to enhanced diffusive upward moisture transport, particularly in regions of strongly deformed zonal flow.

5  Overall, small-scale mixing in the CLaMS simulations and related diffusive cross-tropopause moisture transport seem crucial for the development of Asian and American monsoon moisture anomalies, and in particular for the American monsoon (where no anomaly occurs without including small-scale mixing).

## 4  Discussion

### 4.1  Comparison of CLaMS simulated and reanalysis water vapour

10  The comparison between the reanalysis own specific humidity products and $H_2O$ simulated with CLaMS driven by the meteorology of the same reanalysis reveals further insights into the control processes of $H_2O$ in the reanalysis. Figure 17 shows $H_2O$

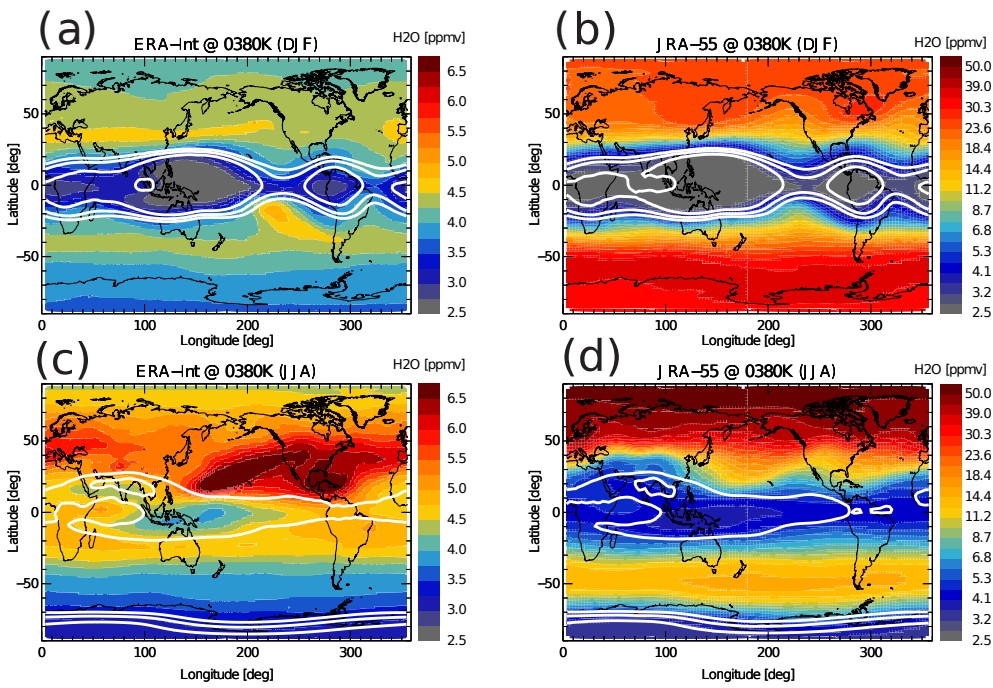

**Figure 17.** Seasonal mean water vapour distribution averaged over the period from 2004 to 2013 at the potential temperature level of 380 K; DJF indicates winter (a, b), and JJA summer periods (c, d). Shown data are from both reanalysis, ERA-Interim and JRA-55 reanalysis data. The white lines are constant temperature levels from corresponding reanalysis datasets (191 K, 193 K, 195 K). Note the logarithmic colorbar for JRA-55.

mixing ratios in the LS at 380 K for winter and summer, as provided by ERA-Interim and JRA-55 specific humidity. Although the two CLaMS simulations driven by either ERA-Interim or JRA-55 showed differences in the details of the patterns (Fig. 5), both simulations agreed reliably well with the satellite observations. The reanalysis $H_2O$ products, on the other hand, show a very different pattern (Fig. 17). Despite their success in describing the main dehydration regions in the deep tropics (mainly in

5   the West Pacific, and over South America in boreal winter), they fail in representing the main moisture sources in the Asian and American monsoons during summer. ERA-Interim, for instance, shows highest summertime $H_2O$ mixing ratios above the Pacific.

The clearest difference to MLS and CLaMS, however, occurs for JRA-55 $H_2O$ in the middle and high latitude LS. In this region, JRA-55 $H_2O$ mixing ratios are about one order of magnitude higher than ERA-Interim. A similar result was recently

10   noticed by Davis et al. (2017), where they showed that JRA-55 strongly overestimates the amplitude of the seasonal $H_2O$ cycle, although this result depends on the considered level. Remarkably, using the reanalysis temperature and wind fields to drive CLaMS transport, results in good agreement of $H_2O$ distributions with MLS observations. Note, that in CLaMS the calculation

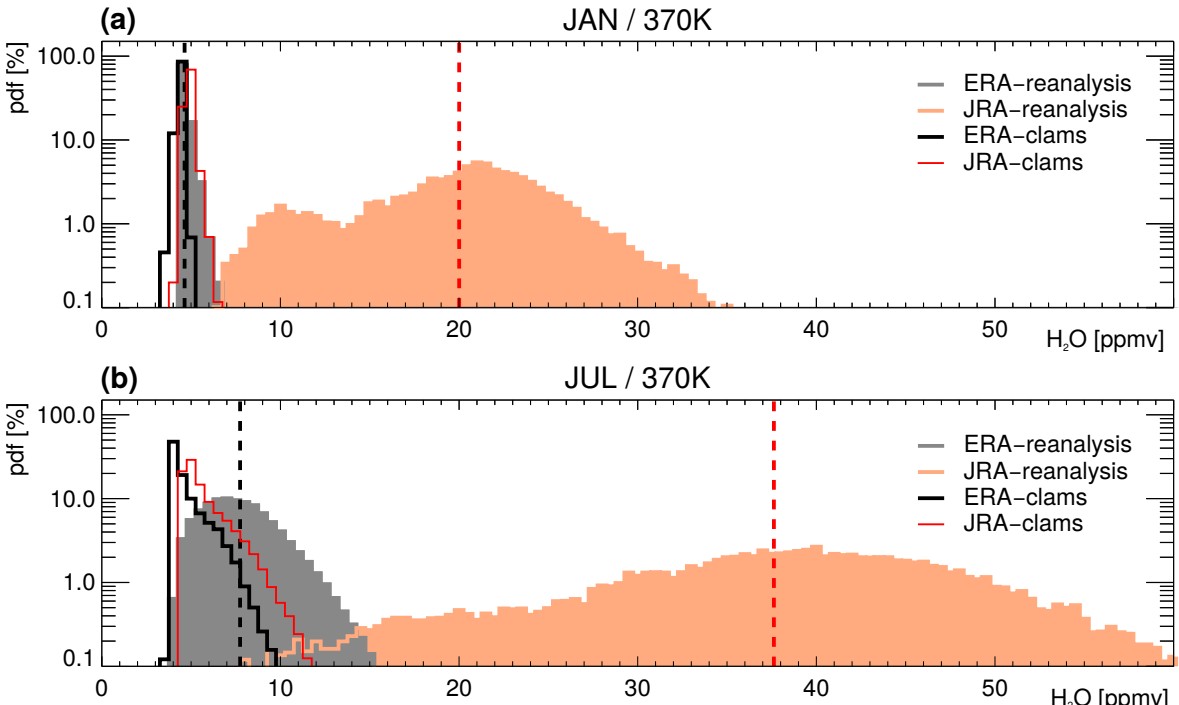

**Figure 18.** Probability density function (PDF) for water vapour mixing ratio for 2011, at the potential temperature level of 370 K in the Northern hemisphere for middle and high latitudes (50°-90° N). The distribution is presented for January (a) and July (b). Shown data are taken from ERA-Interim (grey line) and JRA-55 (orange line) reanalysis data. CLaMS water vapour driven by ERA-Interim (black line) and JRA-55 (red line) is shown for comparison. Vertical dashed lines are the mean values of the reanalysis data respectively, black one of ERA-Interim, red one is JRA-55 reanalysis products.

of stratospheric $H_2O$ is based on the CLaMS cirrus dehydration scheme based on the Clausius-Clapeyron relation and simplified fall-out of ice particles (see Sect. 2) and is, therefore, largely related to the large-scale reanalysis temperature and wind fields. The different CLaMS simulations also show a moister stratosphere for JRA-55, compared to ERA-Interim, consistent with a warmer tropical tropopause in JRA-55, but with much smaller differences than for the reanalysis $H_2O$ products. Although both

reanalysis assimilation systems are constrained with observational data to produce realistic temperatures, significant differences around the tropical tropopause still exist of about 2 K (see Sect. 3.1). But these differences are not sufficient to explain the water vapour differences between the reanalysis products, as presented in Figure 17. Hence, the reanalysis own $H_2O$ products seem not consistent with the simple Clausius-Clapeyron relation.

Figure 18 shows $H_2O$ PDFs for further insights into the processes causing the difference between ERA-Interim and JRA-

55 $H_2O$ in the Northern extratropical LS. Both CLaMS simulations, driven with either ERA-Interim or JRA-55 data, and ERA-Interim reanalysis $H_2O$ products show a skewed PDF with a tail at high values, particularly strong in boreal summer. JRA-55 $H_2O$ mixing ratios, on the contrary, show a PDF with a totally different shape and a much higher mean value by about a factor of 5. This behaviour is the clearest at around 370 K (Fig. 18), but it is also visible at levels below and above (not shown). The different shape of the JRA-55 PDFs, with the peak at much higher mixing ratios, suggests that high $H_2O$

mixing ratios are deposited in the extratropical LS, from potential temperature levels of about 350 K up to at least about 400 K, potentially related to the convective scheme in the reanalysis. JRA-55 shows a higher frequency of high and optically thick clouds, when compared with ERA-Interim (e.g., Kang and Ahn, 2015; Kobayashi et al., 2015; Tompkins et al., 2007), which also could hint at a critical role of differences in convection for causing the differences in $H_2O$ (e.g., Folkins and Martin, 2005; Sherwood et al., 2010).

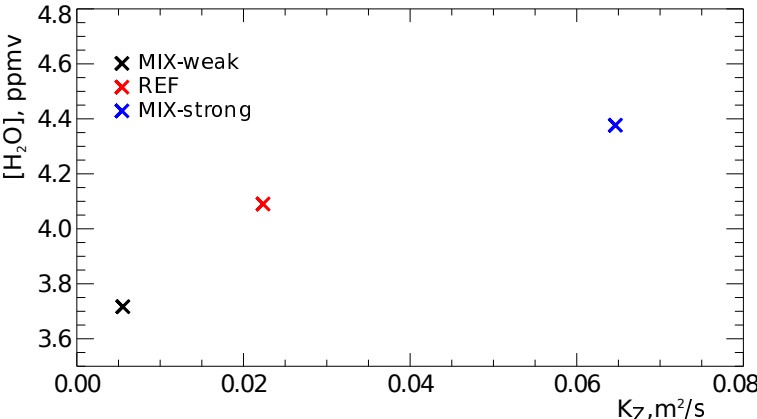

**Figure 19.** Water vapour as a function of the average vertical diffusivity coefficient ($K_z$) in the simulations with different small-scale mixing strength. The data are annual mean values spatially averaged over the 380 K to 420 K potential temperature and 20°S-20°N latitude region from different sensitivity simulations with respect to small-scale mixing (for 2011). Black cross represents weak mixing (MIX-weak, $\lambda_c = 2\,\mathrm{day}^{-1}$), red shows reference (REF, $\lambda_c = 1.5\,\mathrm{day}^{-1}$) and blue cross strong mixing (MIX-strong, $\lambda_c = 1.\,\mathrm{day}^{-1}$) cases.

Recent studies have emphasised the overall qualitatively very good agreement between the large-scale climatological features in the UTLS in different reanalysis datasets, while important quantitative differences remain (e.g., Manney et al., 2017). This qualitative agreement among the reanalysis in many regions of the UTLS and different seasons points to the robustness of the representation of related transport and chemistry in the reanalysis datasets (Manney and Hegglin, 2018). As the stratospheric

$H_2O$ in the reanalysis is not assimilated directly, the treatment of $H_2O$ in the particular reanalysis product plays an important role. For instance, JRA-55 does not contain a parametrisation of methane oxidation in contrary to ERA-Interim (Davis et al., 2017). Davis et al. (2017) further showed that the JRA-55 mean $H_2O$ values are much too large at 100 hPa. Our results of excessively high $H_2O$ values in the JRA-55 data product in the extratropical LS agree well with the findings of Davis et al. (2017). Furthermore, Davis et al. (2017) points out that there is still a lack of assimilated observations and that significant

uncertainties remain in the representation of the relevant physical processes in the reanalysis.

## 4.2    Vertical diffusivity induced by small-scale mixing

Although, it is clear qualitatively that a decreasing critical Lyapunov exponent enhances mixing, it is also desirable, at least for comparison purposes, to quantify this effect. Because of the similarity between the mixing procedure in CLaMS and physical diffusion, the vertical mixing intensity can be quantified by computing the induced vertical diffusivity $K_z$ (in m$^2$/s)

(Konopka et al., 2007). We estimated $K_z$ for each air parcel in CLaMS following Konopka et al. (2007), i.e.:

$$K_z = c \frac{\Delta z^2}{\Delta t} \tag{1}$$

if mixing occurs during $\Delta t$ time step, and $K_z = 0$, if mixing did not happen.    (2)

Here $\Delta z$ is the model layer depth (in meters), $\Delta t = 1$ day is the mixing time step and $c$ is taken to be $1/48$ to account for the random position of the air parcels within the layer. The average diffusivity can be computed as mean of the local diffusivity of

the air parcels.

In the tropical LS (between 380 K and 420 K), we estimate the vertical diffusivity of the simulations with critical Lyapunov exponent $2.0\,\text{day}^{-1}$ (MIX-weak), $1.5\,\text{day}^{-1}$ (REF), and $1.0\,\text{day}^{-1}$ (MIX-strong) to be respectively $0.005\,m^2/s$, $0.02\,m^2/s$, and $0.065\,m^2/s$ (Fig. 19). This range of about one order of magnitude difference for the parametrised $K_z$ in the tropical LS matches the uncertainty in that coefficient (e.g. Podglajen et al., 2017). Indeed, the value parametrised in the reference run (REF) is

similar to the one estimated by Mote et al. (1998) ($0.02\,m^2/s$) from satellite tracer measurements, or to results from a recent study by Podglajen et al. (2017) based on small-scale in-situ wind measurements. Glanville and Birner (2017) rather suggest an average $K_z$ of $0.08\,m^2/s$, what is closer to the value in the MIX-strong simulations.

Figure 19 also shows the annual average $H_2O$ concentration in the tropical LS as a function of the estimated annual average $K_z$ in the same region. The increase of vertical diffusivity by one order of magnitude, from MIX-weak to MIX-strong, results in

a moistening of $\approx 0.6$ ppmv. This sensitivity is of the same order as suggested by Ueyama et al. (2015), although those authors

varied the diffusivity by 3 orders of magnitude. The discrepancy might be partly due to the non-linear dependence of mean $H_2O$ on diffusivity; although there are only 3 points in Fig 19, the impact of increasing vertical diffusivity appears to saturate, probably due to the limitation of transport effects by dehydration.

Finally, it should be noted that the mixing in CLaMS induces both vertical and horizontal diffusion. However, given the larger vertical gradients of $H_2O$ compared to horizontal gradients in the UTLS, the impact of small-scale horizontal diffusion is assumed to be much smaller than the impact of vertical diffusion, especially in the tropics.

## 5 Conclusions

We investigated the sensitivities of modelling $H_2O$ in the LS region regarding different reanalysis datasets, horizontal transport between tropics and extratropics, and small-scale mixing, using the Lagrangian transport model CLaMS.

Differences in $H_2O$ between model simulations driven by ERA-Interim and JRA-55 reanalysis amount to about 0.5 ppmv throughout the stratosphere. This demonstrates a substantial uncertainty in simulated $H_2O$, even when using the most recent reanalysis products. This uncertainty in simulated $H_2O$ results mainly from differences in temperatures between the reanalysis products around the tropical tropopause, indicating that tropopause temperatures in the current reanalysis datasets are not sufficiently constrained.

Sensitivity simulations with introduced artificial transport barriers in the model to suppress certain horizontal transport pathways, shows that the overall effects of interhemispheric transport is weak and insignificant for stratospheric $H_2O$. Furthermore, our results suggest that the NH subtropics are a critical source region of moisture for the global stratosphere, likely related to the subtropical monsoon circulations. Comparison of the tropical entry $H_2O$ from the sensitivity $15° N/S$ barrier run and the reference case shows differences of up to around 1 ppmv. Hence, a reliable representation of processes in the subtropics in global models turns out to be critical for simulating stratospheric $H_2O$ and its climate effects.

Changing the strength of small-scale mixing in CLaMS shows that increased mixing causes moistening of the stratosphere, by enhanced diffusive moisture transport across the tropopause. For the sensitivity simulation with varied mixing strength differences in tropical entry $H_2O$ between the weak and strong mixing cases amount to about 1 ppmv, with small-scale mixing enhancing $H_2O$ in the LS. Interestingly, the impacts of the horizontal transport processes and small-scale mixing are of the same order of magnitude. The strongest mixing effects occur around the subtropical jets in the respective summer hemisphere. In particular, the Asian and American monsoon systems during boreal summer turn out as regions especially sensitive to changes in small-scale mixing, which appears crucial for controlling the moisture anomalies in the monsoon UTLS. Above about 430 K, increased mixing causes a complex interplay between vertical and horizontal mixing, which results in both moistening or drying of the stratosphere depending on the mixing strength. Therefore, interpretation of differences in simulated water vapour from different models in terms of differences in numerical diffusion is a challenging task. The results from our sensitivity simulations are aimed to guide the interpretation of model differences and to pinpoint model deficits.

## Appendix A: Validation of the simulations

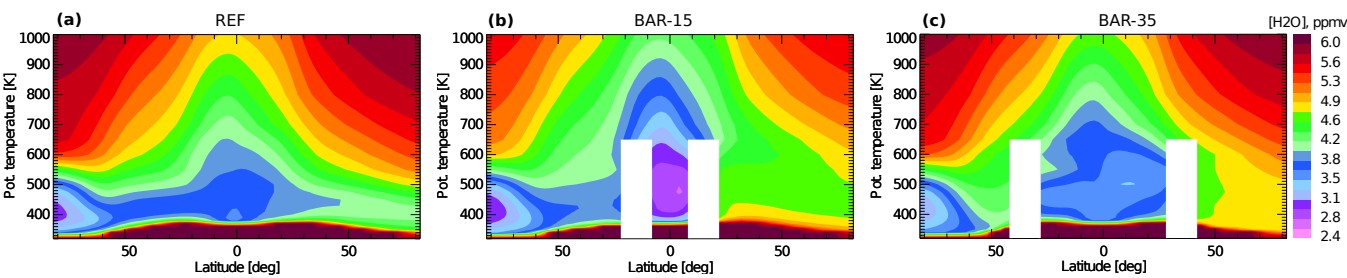

**Figure A1.** Zonal mean water vapour distribution averaged over the period from 2011 to 2014. Shown data are from CLaMS sensitivity simulations for the reference (a) and the horizontal transport barrier simulations with barriers along 15° N/S (b) and 35° N/S (c). Barriers are set between the Earth's surface and the 600 K potential temperature levels and are represented in white.

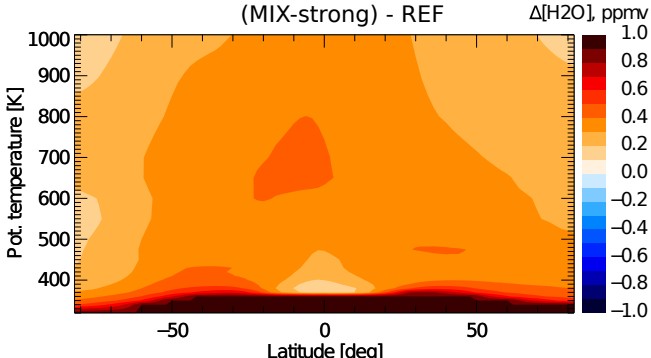

**Figure A2.** Differences of zonal mean water vapour from CLaMS sensitivity simulations with non-vanishing mixing strength between reference (REF) and strong mixing (MIX-strong) cases averaged over the period from 2011 to 2014.

In order to study the robustness of our conclusions concerning changes in the simulation period, we carried out some of the simulations for the entire period between 2011 and 2014. Figure A1 shows the distribution of zonal mean $H_2O$ mixing ratios for this period for the reference (a) and horizontal barrier simulations with barriers at 15°N/S (b) and 35°N/S (c). Clearly, the
5 differences between the different simulations with transport barriers stay qualitatively similar, when compared to the 2011 case (see Fig. 8). Although, the results for the year 2011 appear slightly drier when compared to 2011-2014, likely related to the occurrence of La Niña in 2011.

Also the effects of increased small-scale mixing for 2011 are consistent with the climatological data for the 2011-2014 period (Fig. A2). Although, the effect of increased mixing strength appears even stronger for 2011-2014 (see Fig. 13d for

comparison). As a conclusion, restricting our analysis to a single year has no significant effect on our conclusions, which can be regarded as representative for the climatological case.

*Acknowledgements.* We thank Jens-Uwe Grooß for the helpful discussion. Also we are very appreciative of the ECMWF for providing reanalysis data, the MLS and ACE-FTS teams for providing satellite observation data. The Atmospheric Chemistry Experiment (ACE), also
5   known as SCISAT, is a Canadian-led mission mainly supported by the Canadian Space Agency and the Natural Sciences and Engineering Research Council of Canada. In addition, we gratefully acknowledge the computing time granted on the supercomputer JURECA at Jülich Supercomputing Centre (JSC) under the VSR project ID JICG11. This work was partly funded by the German Ministry of Education and Research under grant no. 01LG1222A (ROMIC-TRIP), and partly by the Helmholtz Young Investigators Group A-SPECi ("Assessment of stratospheric processes and their effects on climate variability").

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
