# Peer review of "Sensitivities of modelled water vapour in the lower stratosphere: temperature uncertainty, effects of horizontal transport and small-scale mixing"

_Atmospheric Chemistry and Physics, 2017_

## Short Comment (SC1) · 30 Nov 2017

The authors discuss transport and feedback processes that modulate stratospheric water vapour changes and the related uncertainties. Could the authors discuss the potential importance of ozone in this context both for the seasonal cycle and climate change projections, see for example:

Fueglistaler, S., Haynes, P. H., and Forster, P. M.: The annual cycle in lower stratospheric temperatures revisited, Atmos. Chem. Phys., 11, 3701-3711,

https://doi.org/10.5194/acp-11-3701-2011, 2011.

Nowack, P. J., Luke Abraham, N., Maycock, A. C., Braesicke, P., Gregory, J. M., Joshi, M. M., Osprey, A., Pyle, J. A.: A large ozone-circulation feedback and its implications for global warming assessments, Nature Climate Change 5, 41-45, https://doi.org/10.1038/nclimate2451, 2015.

Nowack, P. J., Braesicke, P., Abraham, N. L., & Pyle, J. A.: On the role of ozone feedback in the ENSO amplitude response under global warming. Geophysical Research Letters 44, https://doi.org/10.1002/2016GL072418, 2017.

Dietmüller, S., Ponater, M., & Sausen, R. (2014). Interactive ozone induces a negative feedback in CO2-driven climate change simulations. Journal of Geophysical Research: Atmospheres 119, 1796-1805. https://doi.org/10.1002/2013JD020575, 2014.

Could the authors extend their discussion to include the implications of their transport and small-scale mixing work for trace gases more generally, in particular for the distribution of ozone in the TTL, which will feedback onto stratospheric water vapour?

This would be a valuable addition to the manuscript and broaden the potential impact across this interdisciplinary field of research.

---

## Referee Comment (RC1) · R. Ueyama (Referee) · 21 Dec 2017

**General Comments**

This discussion paper addresses the impact of two important yet poorly understood processes controlling water vapor in the UTLS: horizontal transport and small-scale mixing. The CLaMS model is well suited to study this problem. The results help to constrain parameterizations of these processes in climate models, and thus the scientific merit of this paper is significant. My main criticism concerns the somewhat confus-

ing interpretation of the transport barrier simulation results in relation to other metrics such as age of air. The paper would greatly benefit from clearer descriptions and interpretations of the results, aided by improved visualization of some of the figures. I also strongly suggest that the paper focus on the impacts of horizontal transport and small-scale mixing only, which are the novel aspects of this study. The fact that different reanalysis datasets with different tropical tropopause temperatures yield different water vapor results is well known and not particularly interesting. In my opinion, this paper is suitable for publication in Atmospheric Chemistry and Physics after consideration of the specific comments and suggestions provided in detail below.

Major Specific Comments

1. This study describes the effects of three processes - dehydration controlled by tropical tropopause temperature, horizontal transport, and small-scale mixing - on stratospheric H2O. While the results of the latter two processes are novel and interesting, the impact of tropical tropopause temperatures is well known and distracts from the overall significance of this study. Therefore, I strongly suggest that the study focus on the impacts of horizontal transport and small-scale mixing only. There have been many studies that have carefully examined this topic: trajectory modeling sensitivity to temperatures (e.g., Wang et al., 2015), comparison between reanalysis and observed (aircraft, radiosonde) temperatures in the TTL and effect on TTL H2O (e.g., Ueyama et al., 2014, 2015), impact of waves on temperature and dehydration in the TTL (e.g., Jensen and Pfister, 2004; Kim and Alexander, 2015). At least some of these literatures should be referenced to more accurately reflect our current understanding of this issue: while H2O simulations are highly sensitive to tropical tropopause temperatures that still come with some uncertainty, we have a better handle on the accuracy of these temperatures than this paper implies. You may also want to check S-RIP activities and reports (https://s-rip.ees.hokudai.ac.jp/index.html) that discuss the temperature differences between the reanalysis products.

2. The discussion of the effects of the transport barriers (p12-14) could be improved. a.

Specifically, please explain how you are able to discern the direction of transport from these experiments. I understand that the PDFs (Fig. 6) provide a clue, but it is difficult to interpret the relative importance of the two-way transports from these graphs. For example, the barrier at 15N/S makes the tropics drier, which is interpreted as the the lack of equatorward transport of moist air from the subtropics. However, the PDFs in Fig. 6b indicate that there are also more low $H_2O$ mixing ratios in the tropics in the BAR-15 experiment (not just a decrease in high $H_2O$ mixing ratios). Therefore, this suggests that the drier tropics in BAR-15 simulation is also due to the dry air not being transported out of the tropics towards the higher latitudes. Please clarify these points. b. Also, the relationship to the AoA results seems inconsistent at times. The barrier at 35N makes NH extratropics slightly wetter (Fig. 6c), which is explained as a result of the lack of poleward transport of high $H_2O$ mixing ratios from the subtropics. The tropical mean $H_2O$ is unaffected. On the other hand, the barrier at 35N decreases the AoA globally (Fig. 7b), which suggests the importance of the recirculation of aged air from the extratropics into the tropics. Perhaps I'm confusing recirculation with transport, and thus a brief explanation of these terminology in the context of Fig. 11 would be beneficial. c. Please also clarify what you mean on p14, L1-2: "The fact that his drying occurs only with ...." d. The second paragraph of p14 states that horizontal transport exports dry (moist) air out of the tropics in the winter (summer). This is followed by a statement that "the entire annual cycle of LS $H_2O$ in the NH extratropics is related to horizontal transport out of the subtropics". There is some disconnect between these two statements that need to be clarified.

3. Please reconcile the seemingly contradicting results between Fig. 6c vs. Figs. 8acd and Figs. 9acd. Fig. 6c suggests that the lower stratosphere in the NH extratropics is wetter with BAR-15 than with BAR-35, but the opposite seems to be true in Figs. 8 and 9.

4. p18, L3 - p19, L8: The descriptions and explanations provided in this section are confusing. For example, although it states (on p18, L3-4) that a clear response to

mixing is found in the lower stratosphere below 430K (i.e., moistening with increased small-scale mixing), I see significant drying in the UTLS ~350K with the addition of weak small-scale mixing in Figs. 12b and f. Also, on p18, L15, it states that an analogous (drying) signal is found in methane (Fig. 13b), but the figure shows moistening?

5. In the description about small-scale mixing in the CLaMS model, the authors cite the Riese et al. (2012) study. Please include a more detailed explanation of how realistic this mixing parameterization might be, and how it relates to observations. I am assuming that the small-scale mixing refers to both horizontal and vertical diffusion. A recent study by Podglajen et al. (2017) quantified the magnitude of vertical diffusion in the TTL using aircraft observations and found the average diffusivity to be about 0.1 m2s-1 with a strong vertical gradient. Ueyama et al. (2015) found that the "an increase in vertical diffusivity coefficient by 2 orders of magnitude (from 0.001 to 0.1 m2s-1) moistens near the cold point tropopause by approximately 0.5 ppmv". How do your results based on your small-scale mixing parameterization compare to these results? Also, on p6 L15, why are there two numbers for the Lyapunov coefficients that lead to good agreement between observations and simulations?

6. Some of the terminology needs to be better described. For example, please define what you mean by stratospheric H2O (e.g., p14, L1: "global stratosphere becomes substantially drier (up to about 1 ppmv)". Is this a vertically integrated quantity averaged from 90S to 90N? Also, the term "subtropics" is used rather liberally. Sometimes it refers to barriers at 15N/S and other times it refers to the barriers at 35N/S, which can be very confusing. Please clarify your definition and be consistent throughout the text. Along the same line, the barrier is sometimes defined at a specific latitude (e.g., "at 15S/S", p14 L2) and other times within a range of latitudes (e.g., "between 10 and 30 S/N", p13 L23)

7. The Introduction lacks a discussion on the role of convection in lower stratospheric (or TTL) H2O. It is briefly mentioned in the Discussion Section as a possible explanation for the biases in JRA-55 simulations, but convection is an important feature that

needs to be discussed early on. The role of convectively-injected ice is discussed in the Introduction, but its relative importance on stratospheric humidity is still a topic of debate.

8. Figures 6 and 10: The vertical lines are very difficult to see because of the size of the plots and the overlapping lines (e.g., black solid line in panel c?). Vertically stacked panels like Fig. 17 are much easier to see.

Minor Specific Comments

1. It would be helpful to provide the approximate magnitudes ($\sim$x ppmv) of the effects of horizontal mixing and small-scale mixing on stratospheric $H_2O$ in the Abstract and Conclusions (e.g., in the sentence at the top of p22, "Furthermore, our results suggest that the NH subtropics are a critical source region of moisture for the global stratosphere, . . ."). It is worth pointing out that the impacts of these processes are of same order of magnitude as the impact of temperature difference of $\sim$2K.

2. Dessler et al. (2013) paper may not be the most appropriate reference for the first sentence of the Introduction, unless you're specifically emphasizing the feedback effects of stratospheric water vapor. In addition to the papers by Forster and Shine, I would consider the Shindell (2001) paper instead.

3. p 2, L13-14: In your discussion of the rapid transport from the tropics to mid latitudes above the subtropical jets, are you distinguishing between transports in the "tropically controlled transition region" vs. the shallow branch of the Brewer-Dobson circulation, as in Rosenlof et al. (1997)?

4. p2, L15-19: You may want to clarify that the dehydration occurs due to the nucleation and sedimentation of ice crystals, which in essence is a microphysical process controlled by TTL temperatures. Hardiman et al. (2015), which is a study based on global climate models that have difficulty simulating cloud microphysical processes, is not the most appropriate reference here. I would consider citing Jensen et al. (2004,

2005, 2012) studies on the modeling of detailed cloud microphysical processes. Several papers have examined the effect of cloud microphysical processes on humidity of the TTL and stratosphere using cloud models of varying complexity (e.g., Schoeberl et al., 2014; Ueyama et al., 2015).

5. p3, L3-6: I see now that the discussion of temperature and freeze-drying process appears here. I would move this ("The tropical stratospheric entry $H_2O$ mixing ratios ... temperature and vertical velocity fields.") after the paragraph on TTL transport processes in the previous page.

6. p3, L8-10: What exactly are you referring to with respect to the relationship between summer max tropical $H_2O$ and the monsoons? Mixing processes? Deep convection in the monsoon regions? Relatively high tropopause heights?

7. p 3, L18-24: The discussion of TWV here seems a bit out of place. Could you better tie this to the overall discussion about processes that affect lower stratospheric $H_2O$?

8. p5, L20: In the sentence, "Thus, we use the fifth year...for our further analysis", make it clear that you are referring to the analysis of the sensitivity simulations.

9. Table 2: What is "PR"? Would a graph be better for illustrating your point?

10. p7, L1: Is "10-15%" an annual mean value?

11. Figure 2: What is the reason for showing the annual cycle at 400K? What about the 380K level (Figs. 5 and 16 are based on this level)? A brief statement of the sensitivity of your results to the different potential temperatures in the lower stratosphere may be useful. Also, is the MLS averaging kernel applied to the model data for a more accurate comparison to MLS data?

12. p8, L9-10: "The impact of horizontal transport....mixing strength covered." Effects look very similar to me. I would suggest removing this statement to avoid confusion.

13. Figure 5: There are four temperature contours in panels a and b, but only three

contour levels are mentioned in the caption. Devise a better way to label these temperature contours?

14. p11, L7: Please be more specific about what you mean by "agree slightly better" (quantify).

15. p11, L14: I think you mean to say "... show that the interhemispheric transport is rather unimportant..."

16. p13, L11: Where is the "double peak structure"?

17. Figures 12 and 13: Why are these two figures separated? It makes more sense to combine the two and make a 4 x 4 panel figure: TWV, H2O, 2CH4, AoA. Does the sum of Fig. 12a and Fig. 13b equal Fig. 12e?

18. Figure 15: The lines in these figures are very difficult to see.

19. p20, top: It appears that during boreal summer, weak mixing moistens the Asian monsoon region while very strong mixing moistens the N American monsoon region. Do you have an explanation for this interesting result?

20. p20, L30-: Where do you get "380K"? The explanation you provide is certainly plausible, but do these biases occur in mid and high latitudes where the H2O biases are observed?

21. Figure 17: It would be helpful to add MLS data to these plots.

Technical Corrections

1. p6, L15: "....our study, ..."

2. p8, L7: "... tropical (10S-10N) entry of H2O ..."

3. p8, L14: Rewrite this sentence "... reaches the values of ~0.8 ppmv" as "... drying of ~0.8 ppmv in (month)".

4. p10, L17: "... LS H2O between MLS and ClaMS simulations ..."

5. p11, L13; p12, L1: I'm slightly confused by the use of the term "vanishing difference". Are the differences decreasing over time?

6. p 12, L3: "... barrier at 15 N/S changes the PDF ..."

7. p12, L5: "... (blue line in Fig 6b), ..."

8. p12, L7: Rearrange the sentence: "equatorial transport barrier has no effect and cross-equatorial transport is unimportant"

9. p14, L9: Replace with "...very weak drying effect on..."

10. p16, L5: I'm not sure what you mean by "Therefore"

11. p16, L12: " H2O (a-d) and total water (e-h)"

12. p20, L18: "On the contrary, ..." is a very long sentence.

References

Jensen, E., and L. Pfister (2004): Transport and freeze-drying in the tropical tropopause layer.

Jensen, E., et al. (2005): Formation of a tropopause cirrus layer observed over Florida during CRYSTAL-FACE

Jensen, E., et al. (2012): Physical processes controlling ice concentrations in cold cirrus near the tropical tropopause

Kim, J.-E., and J. Alexander (2015): Direct impacts of waves on tropical cold point tropopause temperature.

Podglajen, A., et al. (2017): Small-scale wind fluctuations in the tropical tropopause layer from aircraft measurements: impact on vertical mixing and relationship with clouds and convection.

Schoeberl, M. R., et al. (2014): Cloud formation, convection, and stratospheric dehydration.

Shindell, D. T. (2001): Climate and ozone response to increased stratospheric water vapor.

Ueyama, R., et al. (2014): Dehydration in the tropical tropopause layer: A case study for model evaluation using aircraft observations.

Ueyama, R., et al. (2015): Dynamical, convective, and microphysical control on wintertime distributions of water vapor and clouds in the tropical tropopause layer.

Wang, T., et al. (2014): The impact of temperature vertical structure on trajectory modeling of stratospheric water vapor.

---

## Referee Comment (RC2) · Anonymous Referee #2 · 14 Jan 2018

The paper by L. Poshyvailo et al. tests the importance of three processes (tropopause temperature, horizontal transport and small-scale mixing) for lower stratospheric water vapour concentrations. The paper is overall well written and could be a valuable addition to our understanding and modelling of the tropical upper troposphere and lower stratosphere (UTLS). However, beforehand I recommend several minor revisions.

First of all, I disagree with Reviewer 1 that the discussion on the reanalysis datasets/tropopause temperature should necessarily be entirely removed. While not being a particularly novel part of the paper, it consolidates some previous results and,

in my opinion, helps to put the significance of the other results presented here into context. However, it would be good to add some recent citations for studies which compared reanalysis datasets in the UTLS region, for example:

Manney et al. Reanalysis comparison of upper tropospheric-lower stratospheric jets and multiple tropopauses. Atmos. Chem. Phys., 17, 11541–11566, 2017.

Davis et al. Assessment of upper tropospheric and stratospheric water vapor and ozone in reanalyses as part of S-RIP, Atmos. Chem. Phys., 17, 12743-12778, 2017. (already cited very late in the paper in section 4, but should be mentioned earlier)

Manney & Hegglin. Seasonal and regional variations of long-term changes in upper-tropospheric jets from reanalysis. J. Clim, 31, 423-448, 2018.

A brief discussion on the shortcomings of reanalysis datasets in the tropical UTLS should also be added to the text and could be merged with a shortened discussion (see below) on these issues in section 4. In addition, by referring to these references could help to write some sections (e.g. section 3.1) more concisely.

Specific comments:

page 1:

l. 11: Are JRA-55 and ERA-Interim sufficient to constrain the space of possibilities given by current reanalysis datasets?

l. 12: cancel second mention of 'JRA-55 reanalysis'

l. 12-15: since you mention the difference due to uncertainties in tropopause temperatures (0.5 ppmv), could you provide more quantitative statements for the other two processes?

l. 21: reference to Nowack et al. (2015) could be added here.

Nowack et al. A large ozone-circulation feedback and its implications for global warming assessments, Nature Climate Change 5, 41-45, 2015.

l. 23: reference to Maycock et al. was published in Journal of Climate, see typo in reference list. Reference to Nowack et al. (2017) could be added here for another recent example of how UTLS processes/stratospheric water vapor can influence climate variability:

Nowack et al. On the role of ozone feedback in the ENSO amplitude response under global warming. Geophys. Res. Lett. 44, 3858-3866, 2017.

page 2:

l. 12/13: you cite studies about the importance of the Asian Monsoon below. Would add those references here already.

l. 16: to avoid confusion with too many 'lows' I recommend to change to: 'freeze-dried to stratospheric values', which is self-explanatory.

l. 19: next to Hardiman, 2015, you could add Schoeberl et al. 2014 here:

Schoeberl et al. Cloud formation, convection, and stratospheric dehydration, Earth and Space Science 1, 1–17, 2014.

page 3:

l. 5: reanalysis,

l. 6: delete 'largely'

l. 7-11: I agree that the lack of discussion on ozone as a process in the introduction is a shortcoming of the current manuscript. In this paragraph, there would be an opportunity to mention the importance of ozone in modulating stratospheric water vapor concentrations. In the discussion section, ozone could be included as a potential future research interest, because ozone will equally be transported by mixing etc. An ozone-focused analysis is beyond the scope of this study though, so not necessary,

but a brief discussion would be useful.

l. 25: Based on model simulations,

l. 26: can cause

l. 33: change to: 'with respect to two meteorological datasets...', following sentences add references to the recent studies on reanalysis data indicated above.

page 4:

l. 27: about 2 million

l. 31: given the importance of the parameter for the simulations here, a somewhat more detailed description would be helpful rather than just referring to other papers.

page 5:

l. 5: the naming convention here implies that this pcold_point is the ambient pressure in the tropopause, but this does not seem to fit your explanation? Either be more specific about the ambient pressure or change the naming of the parameter.

page 7:

l. 3-5: in Table 1, the no mixing experiment is labelled with a mixing parameter of 0, whereas here you say that larger values represent weaker mixing. How does this fit together?

l. 5/6: it is however unclear how non-linear the effects of mixing scale with the mixing strength. This should at least be pointed out, or alternatively could be tested by running additional simulations with intermediate size parameters.

page 8:

l. 13/14: These values should also be given above, especially in the abstract.

Figure 2: x-axis, label DJFM..., or Dec Mar...currently it is unclear if you start in January? December?

l. 15: good opportunity to cite other studies that have looked at this before. Any differences?

page 9:

Figure 3: Remove space between the second and third month of each season's label. I think it would also be better to plot the MLS climatology in the first row and differences relative to that in the lower three rows. This would make the structure of the differences more obvious.

page 10:

l. 2: no comma after detail

l. 10: either...or

page 11:

Figure 5: revise titles of the subfigures. Again, I would recommend plotting differences for (b), (c), (e) and (f), especially given that the rainbow color scale skews the perspective on the maps, while the color choice is also not ideal (colorblind readers). I recommend choosing a different color scale.

page 12:

l. 5: skewed towards...

Fig. 6(a)-(c) should be reordered to match the order of the discussion in the text, or at least NH and SH swapped.

page 13:

Figure 7: again I would replace the color scale by something gradual, e.g. cold-to-warm color scale. Currently, the differences are really hard to see. Plotting differences in (b) and (c) would help, too. Maybe use contour lines for REF in order to use a single

color scale?

l. 7: typo reference

l. 14: Why is there a consequent increase in age of air in the global stratosphere if the age is increased in the extratropics?

page 14:

Figure 8: again difference plots relative to REF would be better, same changes as above concerning the color scale.

l. 10: simulations typo

page 18:

l. 9 typo

Figure 14: Clarify that these are percentage differences in the figure caption

page 19:

Figure 15: again, I would suggest a change of color scale, the plots are very hard to read. In addition, change (b,c,d,f,g,h) to difference plots relative to MIX-no. Too much information for a single plot due to the two types of contour lines.

page 20:

l. 10: reliably well, or maybe fairly well or just 'well'?

page 21:

Figure 16: revise titles

page 22:

Figure 17 (and the corresponding text): I agree with Reviewer 1 that this part of the reanalysis discussion could be kept much shorter. Issues around stratospheric water

vapor are known for reanalysis datasets as the authors point themselves out in the text. Discussing this topic again here in so much detail and with an extra figure distracts from the key messages of the paper, i.e. how the three sub-processes influence stratospheric water vapor concentrations.

---

## Author Comment (AC1) · 23 Mar 2018

We thank P. J. Nowack for the helpful comment. We give a point-by-point reply below, where the reviewer comments are repeated in italics. The positions of the corrected sentences in the revised version are noted in the brackets, and the revised text is also given in the quotation marks point-by-point below.

[Figure]

Specific Comments

1. *The authors discuss transport and feedback processes that modulate strato-spheric water vapour changes and the related uncertainties. Could the authors discuss the potential importance of ozone in this context both for the seasonal cycle and climate change projections...*

   Thank you for this suggestion. Although this paper focusses on processes controlling stratospheric $H_2O$, we added a brief discussion on stratospheric ozone (p3, L21).

   "...Furthermore, it has been pointed out that the coupling between ozone, the tropospheric circulation, and climate variability plays an important role in climate change (Nowack et al., 2017). Recent studies have shown that stratospheric ozone changes may cause an increase in global mean surface warming, mostly induced by changes in long-wave radiative feedbacks due to the tropical LS ozone and related stratospheric $H_2O$ and cirrus cloud changes (e.g., Nowack et al., 2015; Dietmüller et al., 2014). Seasonal variations of LS ozone lead to a magnification of the seasonal temperature cycle in the tropics (Fueglistaler et al., 2011). Investigation of these additional effects of stratospheric ozone is an important topic of future research focussed on stratospheric $H_2O$ feedbacks..."

2. *Could the authors extend their discussion to include the implications of their transport and small-scale mixing work for trace gases more generally, in particular for the distribution of ozone in the TTL, which will feedback onto stratospheric water vapour?*

   Thank you for this question. It is answered partially above. As an ozone-focused analysis is beyond the scope of this study, we do not think that it is necessary to extend the discussion of the transport and small-scale mixing effects to other trace gases more generally (as the paper is focused on stratospheric $H_2O$). An assessment of feedbacks from ozone on $H_2O$ is unfortunately not feasible in a

pure transport model like CLaMS. We are working on coupling CLaMS into the climate model EMAC to enable such analysis, but this development is beyond the scope of this paper.

**References**

Dietmüller, S., Ponater, M., and Sausen, R.: Interactive ozone induces a negative feedback in CO2-driven climate change simulations, Journal of Geophysical Research: Atmospheres, 119, 1796–1805, doi:10.1002/2013JD020575, URL http://dx.doi.org/10.1002/2013JD020575, 2014.

Fueglistaler, S., Haynes, P. H., and Forster, P. M.: The annual cycle in lower stratospheric temperatures revisited, Atmos. Chem. Phys., 11, 3701–3711, doi:10.5194/acp-11-3701-2011, 2011.

Nowack, P. J., Luke, A. N., Maycock, A. C., Braesicke, P., Gregory, J. M., Joshi, M. M., Osprey, A., and Pyle, J. A.: A large ozone-circulation feed- back and its implications for global warming assessments, Nature Climate Change, 5, 41–45, doi:10.1038/nclimate2451, 2015.

Nowack, P. J., Braesicke, P., Luke Abraham, N., and Pyle, J. A.: On the role of ozone feedback in the ENSO amplitude response under global warming, Geophysical Research Letters, 44, 3858–3866, doi:10.1002/2016GL072418, 2017.

---

## Author Comment (AC2) · 23 Mar 2018

We thank the referee for the detailed review and for very helpful comments. We give a point-by-point reply below, where the reviewer comments are repeated in italics. The positions of the corrected sentences in the revised version are noted in the brackets, and the revised text is also given in the quotation marks point-by-point below.

[Figure]

**General remarks**

*This discussion paper addresses the impact of two important yet poorly understood processes controlling water vapour in the UTLS: horizontal transport and small-scale mixing. The CLaMS model is well suited to study this problem. The results help to constrain parametrizations of these processes in climate models, and thus the scientific merit of this paper is significant... This paper is suitable for publication in ACP after consideration of the specific comments and suggestions provided in detail below.*

Thank you very much for this positive comment. In the revised version all comments have been taken into account, particularly we improved the discussion part, presentation of figures and formulation of the statements throughout the text related to the remarks of the Reviewer #1.

**Major Specific Comments**

1. *...I strongly suggest that the study focus on the impacts of horizontal transport and small-scale mixing only...*

   Thank you for this suggestion. However, Referee #2 pointed out that the discussion of the reanalysis datasets/tropopause temperature is also important and interesting and should not be removed, as *it consolidates some previous results and, in my opinion, helps to put the significance of the other results presented here into context.* So, we prefer to leave the discussion about the reanalysis TTL temperatures and the impact on LS H2O. Also we added some references regarding TTL processes and their impact on LS H2O distribution (p2, L22).

   "...Related to the mean upward transport, the TTL includes the region of very low temperatures around the cold-point tropopause, where the moist tropospheric air is freeze-dried to stratospheric values (Brewer, 1949). Thus, the tropical cold-point temperatures control the amount of $H_2O$, which enters the stratosphere

(e.g., Wang et al., 2015; Kim and Alexander, 2015). The dehydration occurs as a result of the slow upward and large-scale horizontal motion of air in this region (Holton and Gettelman, 2001), where the nucleation and sedimentation of ice crystals take place, which in essence is a microphysical process controlled by TTL temperatures. The freezing is sensitive not just to large-scale TTL temperatures, but also to microphysical processes controlling the ice crystal number densities, particle size distribution, and fall speed. Several studies focused on the modelling of the detailed cloud microphysical processes (e.g., Jensen and Pfister, 2004; Jensen et al., 2005,2012). Other recent papers have examined the effect of cloud microphysical processes on the humidity of the TTL and stratosphere using cloud models of varying complexity..."

2. *The discussion of the effects of the transport barriers (p12-14) could be improved.*

   a *...how you are able to discern the direction of transport from these experiments...*

   We agree that this point was not discussed appropriately enough. Here we changed the order of the description, from Fig.6a to Fig.6c (p12, L13 – p14, L22), and we added an explanation regarding Fig.6b.

   BAR-15 represents suppressed horizontal transport from the subtropics into the tropics and vice verse. Similarly, in-mixing of mid- and high-latitude air and transport from the subtropics to extratropics is presented with BAR-35.

   "...Similarly, in-mixing of mid- and high-latitude air (see BAR-35) has a very small impact on tropical mean $H_2O$, in agreement with the findings of Ploeger et al. (2012). In contrast, transport from the subtropics into the tropics has a strong effect. Suppressing such transport by applying a barrier at $15°$ N/S (BAR-15) changes the PDF substantially, as evident from the difference between the simulation BAR-15 and the reference cases. The isolation of the tropics due to the lack of horizontal transport in the BAR-15 simulation (all the way from the surface to 600 K) between equator and subtropics

(both ways) causes dry air at the equator. Thus, with the barrier at 15° N/S the fraction of dry air at the equatorial region increases. The comparison of BAR-15 with BAR-35 shows that transport from the subtropical region into the tropics increases $H_2O$..."

b *Also, the relationship to the AoA results seems inconsistent at times...*

We agree, that the discussion was to sketchy. So we, added more explanation regarding it (p15, L1).

"...The pure transport effects of horizontal exchange between tropics and mid-latitudes are evident from mean age of air (AoA), the mean transit time for air through the stratosphere for the different model experiments with horizontal transport barriers. Figure 7 shows CLaMS calculations of the AoA for the reference case (Fig. 7a), simulation with transport barriers in the subtropics at 35° N/S (Fig. 7b) and the absolute difference between them (Fig. 7c). These horizontal transport barriers at 35° N/ effectively isolate the tropical pipe from the in-mixing of older stratospheric air from mid-latitudes... A similar result has been recently found by Garny et al. (2014). Furthermore, Garny et al. (2014) presents a nice explanation of the recirculation process, describing recirculation as a process when an air parcel enters the tropical stratosphere and travels along the residual circulation to the extratropics, where it can be mixed back into the tropics, and thus recirculates along the residual circulation again. In this way, the age of air of the parcels increases steadily while performing multiple circuits through the stratosphere..."

c *Please also clarify what you mean on p14, L1-2: "The fact that his drying occurs only with..."*

With both, BAR-15 and BAR-35, recirculation is suppressed, as the region of the extratropics is isolated from the tropics with the barriers. This suppresses the recirculation of extratropical air into the tropics (p15, L28).

"...The fact that this drying occurs only with transport barriers at $15°$ N/S and

not with barriers at $35°$ N/S, shows that it is not related to the suppression of recirculation of aged air from mid-latitudes, which has been affected by methane oxidation. In fact, processes in the subtropics (e.g., monsoon circulations) have a strong effect in moistening the global stratosphere, and suppressing these processes in BAR-15 causes drying. ..."

d *The second paragraph of p14 states that horizontal transport exports dry (moist) air out of the tropics in the winter (summer). This is followed by a statement that "the entire annual cycle of LS H2O in the NH extratropics is related to horizontal transport out of the subtropics". There is some disconnect between these two statements that need to be clarified.*

Thank you for pointing this unclear formulation out. We changed the formulation of the statement (p16, L5). Also we added some changes to the Fig.9, we think this will add more clarity to the explanation.

"...Therefore, during winter, horizontal transport exports dry air out of the tropics into the NH, and, during summer, moist air. Consequently, the entire annual cycle of the $H_2O$ in the NH extratropical LS is related to horizontal transport from low latitudes, as argued by Ploeger et al. (2013). The boreal summer maxima are related to monsoonal circulations and transport out of the tropics, along the eastern and western flanks (Randel and Jensen, 2013)..."

3. *Please reconcile the seemingly contradicting results between Fig. 6c vs. Figs. 8acd and Figs. 9acd. Fig. 6c suggests that the lower stratosphere in the NH extratropics is wetter with BAR-15 than with BAR-35, but the opposite seems to be true in Figs. 8 and 9.*

It is not straightforward to compare Fig. 6c with Figs. 8acd and Figs. 9acd. This is due to the fact that Figs. 8,9 only show the mean value for 2011 while Fig. 6 shows the distribution of all H2O values. Therefore, it is only possible to compare the mean value of H2O from the distributions in Fig. 6 (as presented by the

vertical lines) with Figs. 8acd, 9acd. These values are around 5 ppmv for both BAR-15 and BAR-35, and agree well between all figures so that no contradiction occurs.

4. *p18, L3 - p19, L8: The descriptions and explanations provided in this section are confusing. For example, although it states (on p18, L3-4) that a clear response to mixing is found in the lower stratosphere below 430K (i.e., moistening with increased small-scale mixing), I see significant drying in the UTLS âĹij350K with the addition of weak small-scale mixing in Figs. 12b and f.*

Thank you for pointing this potential for confusion out. Our sensitivity studies are suitable only for the stratosphere, as the simple H2O parametrization in CLaMS is adequate only above the cold-point tropopause. Also we added a discussion of this issue, and reformulated some part of the text (p6, L12; p.19, L.9). In Figs. 12,13 (now it is 13,14) we also show the tropopause for better illustration of the separation of stratosphere from the troposphere.

"...Note, that the CLaMS $H_2O$ calculation gives meaningful results only above the tropopause due to the simple parametrization of ice microphysics and not including a convection parametrization. In the stratosphere, however, CLaMS $H_2O$ has been shown to agree well with the observations (e.g., Ploeger et al., 2013)..."

"...A clear response to mixing is found for the LS (below $\approx 430\,\mathrm{K}$), which is moistened with increasing small-scale mixing. In the following, we consider total water above the tropical tropopause as an indicator of changes in transport because it is not affected by chemistry (here methane oxidation). As the moistening in the LS below $430\,\mathrm{K}$ is also evident in total water, but not in methane and mean age, it is largely related to enhanced diffusive cross-tropopause transport of moist air..."

*Also, on p18, L15, it states that an analogous (drying) signal is found in methane (Fig. 13b), but the figure shows moistening?*

This paragraph discusses the region above 430K, where the air becomes younger and contains more CH4 (in SH/NH in the middle and upper stratosphere).

"...A related signal (above 430K in the middle and upper stratosphere) is evident in methane (Fig. 14b) and mean age (Fig. 14f), but not in total water (Fig. 13f)...."

5. *In the description about small-scale mixing in the CLaMS model, the authors cite the Riese et al. (2012) study. Please include a more detailed explanation of how realistic this mixing parametrization might be, and how it relates to observations.*

Thanks for this suggestion to improve the explanation and discussion of CLaMS mixing. We had included already references to some other papers (Konopka 2004, 2005). In the revised version we extended the explanation of CLaMS mixing and in particular added a discussion about how realistic the mixing parametrization in CLaMS (p.5, L27) is.

"...A validation of the CLaMS mixing scheme was presented by Konopka et al. (2005a) in comparison to CRISTA-1 observations. Importantly, the CLaMS mixing parametrization affects both vertical and horizontal diffusivity. Horizontal diffusivity is largely associated with deformation in the horizontal flow. The vertical mixing is mainly related to the vertical shear (Konopka et al., 2004, 2005b)..."

*I am assuming that the small-scale mixing refers to both horizontal and vertical diffusion.*

Thank you for this comment. And yes, it refers to both horizontal and vertical mixing. This is pointed out more clearly in the revised version (see the answer above).

*A recent study by Podglajen et al. (2017) quantified the magnitude of vertical diffusion in the TTL using aircraft observations and found the average diffusivity to be about 0.1 m2s-1 with a strong vertical gradient. Ueyama et al. (2015)*

*found that the "an increase in vertical diffusivity coefficient by 2 orders of magnitude (from 0.001 to 0.1 m2s-1) moistens near the cold point tropopause by approximately 0.5 ppmv". How do your results based on your small-scale mixing parameterization compare to these results?*

Thank you for this comment. We added information about the connection between small-scale mixing and vertical diffusivity in the revised manuscript. Therefore, we estimated the vertical diffusivity for the different CLaMS sensitivity simulations and provide the numbers in a new Fig. 19 (p.25-27).

"...In addition, we estimated the vertical diffusivity coefficient for the TTL for the different model simulations. The result suggests a non-linear response of H2O to the small-scale mixing in CLaMS (details are considered in Sect. 4)..."

"...Although, it is clear qualitatively that a decreasing critical Lyapunov exponent enhances mixing, it is also desirable, at least for comparison purposes, to quantify this effect. Because of the similarity between the mixing procedure in CLaMS and physical diffusion, the vertical mixing intensity can be quantified by computing the induced vertical diffusivity $K_z$ (in m$^2$/s) (Konopka et al., 2007)... Finally, it should be noted that the mixing in CLaMS induces both vertical and horizontal diffusion. However, given the larger vertical gradients of H$_2$O compared to horizontal gradients in the UTLS, the impact of small-scale horizontal diffusion is assumed to be much smaller than the impact of vertical diffusion, especially in the tropics..."

*...p6 L15, why are there two numbers for the Lyapunov coefficients that lead to good agreement between observations and simulations?*

Thank you for this remark. We added some detailed explanation to the text (p7, L5). And the brief explanation is that these two values, 1.2 and 1.5, describe well the stratospheric behaviour when compared to the observations, depending on whether the model is run in a 2D or 3D set-up (1.2 for 2D, 1.5 for 3D).

"...provide good agreement between observations and simulation results, as found from comparison of CLaMS simulations with observations from infrared limb-sounding from the research aircraft Geophysica (Khosrawi et al., 2005). In particular, using the value of $1.2 \, day^{-1}$ gives a better agreement with observations in the 2D version of CLaMS (Konopka et al., 2003). Furthermore, Konopka et al. (2004, 2005b) showed that the value of $\lambda_c$ = $1.5 \, day^{-1}$ (corresponding to the critical deformation of $\gamma_c = 1.5$) for the chosen horizontal resolution and time step here, turns out to be optimal for the 3D version of CLaMS. Even for such a small difference in the small-scale mixing strengths, annual mean $H_2O$ concentrations in the extratropical LS differs by about 10-15% (Riese et al., 2012; McKenna et al., 2002a). In our study we use a value of $\lambda_c$ = $1.5 \, day^{-1}$ for the reference run, $2.0 \, day^{-1}$ to represent weak mixing, and $1.0 \, day^{-1}$ for modelling strong mixing to cover the range of realistic small-scale mixing strength..."

6. *Some of the terminology needs to be better described. For example, please define what you mean by stratospheric H2O (e.g., p14, L1: "global stratosphere becomes substantially drier (up to about 1 ppmv)". Is this a vertically integrated quantity averaged from 90S to 90N?*

Thank you for these remarks. This explanation refers to the comparison between Figs.8ad. Hence, the statement concerned the maximum local difference. These maximum differences of H2O in the stratosphere can reach up to 1 ppmv.We improved the explanation (p.15, L.27).

"...Without such transport from the subtropics (Fig. 8a, 8d), the stratosphere becomes substantially drier (maximal differences in the stratosphere are up to about 1 ppmv)..."

*Also, the term "subtropics" is used rather liberally. Sometimes it refers to barriers at 15N/S and other times it refers to the barriers at 35N/S, which can be very confusing. Please clarify your definition and be consistent throughout the text.*

We improved the explanation and clarified the use of the BAR-terminology in the revised version (p6, L31). In particular, we explain how both barriers suppress the impact of the subtropics, depending on which transport is considered.

"...The two types of barriers, BAR-15 and BAR-35, are located at the edge of the subtropics. BAR-15 is located at the equatorward edge and BAR-35 at the poleward edge of the subtropics. So, both of them inhibit the transport from the subtropics. BAR-15 suppresses horizontal transport from the subtropics into the tropics, and BAR-35 suppresses transport from the subtropics to the extratropics...."

*Along the same line, the barrier is sometimes defined at a specific latitude (e.g., "at 15S/S", p14 L2) and other times within a range of latitudes (e.g., "between 10 and 30 S/N", p13 L23).*

Thank you for this remark regarding the terminology. The barriers are always 10 degrees in width, centred at 0, 15, 35 degrees. We tried to remove all ambiguous formulations throughout the paper (p6, L29).

"...The transport barriers are defined in the model and centred at the given latitude. Their thickness is $10°$ in latitude (to inhibit diffusive mixing transport), and the barriers extend from the ground to 600 K potential temperature..."

7. *The Introduction lacks a discussion on the role of convection in lower stratospheric (or TTL) H2O. It is briefly mentioned in the Discussion Section as a possible explanation for the biases in JRA-55 simulations, but convection is an important feature that needs to be discussed early on. The role of convectively-injected ice is discussed in the Introduction, but its relative importance on stratospheric humidity is still a topic of debate.*

Thank you for this comment. We agree that convection is an important process controlling stratospheric water vapour. We included a discussion of convection in the introduction in the revised manuscript (p3, L3).

"...Sublimation of ice, injected by deep convection, has also been argued to be an important factor for the H2O budget of the tropical LS (e.g., Avery et al., 2017; Jensen and Pfister, 2004). Convection affects the transport of water and ice and influences the temperatures over the convective region, in turn affecting dehydration (e.g., Fueglistaler et al., 2009). The predominant impact of convection has been shown to moisten the TTL by up to 0.7 ppmv at 100 hPa level and even more below this level (e.g., Ueyama et al., 2014, 2015). Similarly, Schoeberl et al. (2014) argued that an increase of convection will increase stratospheric H2O and tropical cirrus around the cold-point tropopause. At higher levels in the TTL, however, the moistening effect of convection appears very weak (e.g., Schiller et al., 2009)..."

8. *Figures 6 and 10: The vertical lines are very difficult to see because of the size of the plots and the overlapping lines (e.g., black solid line in panel c?). Vertically stacked panels like Fig. 17 are much easier to see.*

   Thank you for this comment. We changed the Figures 6 and 10 (now it is Fig.11) as proposed by the Reviewer.

**Minor Specific Comments**

1. *It would be helpful to provide the approximate magnitudes (âĹijx ppmv) of the effects of horizontal mixing and small-scale mixing on stratospheric H2O in the Abstract and Conclusions (e.g., in the sentence at the top of p22, "Furthermore, our results suggest that the NH subtropics are a critical source region of moisture for the global stratosphere, . . ."). It is worth pointing out that the impacts of these processes are of same order of magnitude as the impact of temperature difference of âĹij2K.*

   Thank you for the comment. We changed the text according to the Reviewer's suggestions in the Abstract and the Conclusions (p1, L14).

"...Comparison of tropical entry $H_2O$ from the sensitivity $15°$ N/S barrier simulation and the reference case shows differences of up to around 1 ppmv... For the sensitivity simulation with varied mixing strength differences in tropical entry $H_2O$ between the weak and strong mixing cases amount to about 1 ppmv, with small-scale mixing enhancing $H_2O$ in the LS...."

2. *Dessler et al. (2013) paper may not be the most appropriate reference for the first sentence of the Introduction, unless you're specifically emphasizing the feedback effects of stratospheric water vapor. In addition to the papers by Forster and Shine, I would consider the Shindell (2001) paper instead.*

We changed the reference in the revised version according to the Reviewer's suggestion (p2, L1).

"...Stratospheric water vapour (H2O) is a crucial factor for global radiation, as it cools the stratosphere and warms the troposphere (e.g., Forster and Shine, 1999, 2002; Shindell, 2001; Nowack et al., 2015)..."

3. *p 2, L13-14: In your discussion of the rapid transport from the tropics to mid latitudes above the subtropical jets, are you distinguishing between transports in the "tropically controlled transition region" vs. the shallow branch of the Brewer-Dobson circulation, as in Rosenlof et al. (1997)?*

Thank you for this comment regarding clarification of the terminology. We do not distinguish between transports in the "tropically controlled transition region" vs. the shallow branch of the Brewer-Dobson circulation. In our opinion both processes are strongly related, and the transport in the tropically controlled transition region may even belong to the shallow branch of the Brewer-Dobson circulation.

4. *p2, L15-19: You may want to clarify that the dehydration occurs due to the nucleation and sedimentation of ice crystals, which in essence is a microphysical process controlled by TTL temperatures. Hardiman et al. (2015), which is a study*

[Figure]

*based on global climate models that have difficulty simulating cloud microphysical processes, is not the most appropriate reference here. I would consider citing Jensen et al. (2004,2005, 2012) studies on the modeling of detailed cloud microphysical processes. Several papers have examined the effect of cloud microphysical processes on humidity of the TTL and stratosphere using cloud models of varying complexity (e.g., Schoeberl et al., 2014; Ueyama et al., 2015).*

Thank you for these clarifications. We changed the text and referencing according to the suggestions (p2, L22).

"...Related to the mean upward transport, the TTL includes the region of very low temperatures around the cold-point tropopause, where the moist tropospheric air is freeze-dried to stratospheric values (Brewer, 1949). Thus, the tropical cold-point temperatures control the amount of $H_2O$, which enters the stratosphere (e.g., Wang et al., 2015; Kim and Alexander, 2015). The dehydration occurs as a result of the slow upward and large-scale horizontal motion of air in this region (Holton and Gettelman, 2001), where the nucleation and sedimentation of ice crystals take place, which in essence is a microphysical process controlled by TTL temperatures. The freezing is sensitive not just to large-scale TTL temperatures, but also to microphysical processes controlling the ice crystal number densities, particle size distribution, and fall speed. Several studies focused on the modelling of the detailed cloud microphysical processes (e.g., Jensen and Pfister, 2004; Jensen et al., 2005, 2012). Other recent papers have examined the effect of cloud microphysical processes on the humidity of the TTL and stratosphere using cloud models of varying complexity (e.g., Ueyama et al., 2015; Schoeberl et al., 2014)..."

5. *p3, L3-6: I see now that the discussion of temperature and freeze-drying process appears here. I would move this ("The tropical stratospheric entry H2O mixing ratios ... temperature and vertical velocity fields.") after the paragraph on TTL transport processes in the previous page.*

We moved it to the paragraph above (p2, L31).

"...The tropical stratospheric entry $H_2O$ mixing ratios can be well simulated by the advection through the large-scale temperature field and instantaneous freezing, often described as the "advection-condensation" paradigm (Pierrehumbert and Rocca, 1998; Fueglistaler and Haynes, 2005)..."

6. *p3, L8-10: What exactly are you referring to with respect to the relationship between summer max tropical H2O and the monsoons? Mixing processes? Deep convection in the monsoon regions? Relatively high tropopause heights?*

This is indeed a good question. The summertime monsoon systems very likely involve all three mentioned processes. The relative strength of these processes, however, is an open question and a topic of current research. Based on out model experiments, we can not estimate which processes dominates in the real atmosphere. We, therefore, rewrote this sentence (p3, L18).

"...The summer maximum of tropical $H_2O$ mixing ratios has been argued to be also related, to some degree, to the subtropical monsoon circulations like the Asian monsoon. However, the strength of this effect and the detailed processes involved (e.g., deep convection, large-scale upwelling) is a matter of debate..."

7. *p 3, L18-24: The discussion of TWV here seems a bit out of place. Could you better tie this to the overall discussion about processes that affect lower stratospheric H2O?*

Thank you for this suggestion. We shifted this sentence to the paragraph above (p3, L10).

"...Above the TTL, $H_2O$ behaves mainly as a tracer, and the tape recorder signal imprinted at the cold-point tropopause ascends deep into the tropical stratosphere (Mote et al., 1996). At higher altitudes in the stratosphere, methane oxidation results in a chemical source for stratospheric $H_2O$ (e.g., LeTexier et

al., 1988; Rohs et al., 2006). As a net result of this oxidation process each methane molecule is converted into approximately two $H_2O$ molecules. Hence, the total water vapour (TWV), TWV $= 2CH_4 + H_2O$, is unchanged by transport in the stratosphere and can be regarded approximately constant (e.g., Dessler et al., 1994; Mote et al., 1998; Randel et al., 1998). Therefore, the sum $2CH_4 + H_2O$ is an important value to indicate the amount of water entering the stratosphere (e.g., Kampfer, 2013). The annual cycle of TTL temperatures (minimum in boreal winter, maximum in summer) is imprinted on $H_2O$ mixing ratios entering the stratosphere, forming the so-called "tape recorder" signal (Mote et al., 1995, 1996)..."

8. *p5, L20: In the sentence, "Thus, we use the fifth year. . .for our further analysis", make it clear that you are referring to the analysis of the sensitivity simulations.*

We rewrote this sentence.

"...Thus, we use the fifth year of the perpetuum simulation for our further analysis..."

9. *Table 2: What is "PR"? Would a graph be better for illustrating your point?*

Thank you for this remark concerning the terminology. PR-files are the files after each year of perpetuum run. We agree that this terminology is not helpful for the readership and that also giving the differences between all the spin-up years provides not too much insight. The necessary information is simply that after the fourth year the maximum changes are below 1%. This information is now given in the revised manuscript and all other unnecessary information is removed (p6, L19).

"...After the fourth year, the maximum relative change of $H_2O$ mixing ratios between further years of the simulation is very small with the defined resolution and the time step (maximum year to year changes are below 1.0 %). Thus, we use the fifth year of the perpetuum simulation for our further analysis..."

10. *p7, L1: Is "10-15%" an annual mean value?*

    Yes, it is an annual mean differences for 2003 (Riese et al. 2012).

    "...Even for such a small difference in the small-scale mixing strengths, annual mean $H_2O$ concentrations in the extratropical LS differs by about 10-15% (Riese et al., 2012; McKenna et al., 2002a)..."

11. *Figure 2: What is the reason for showing the annual cycle at 400K? What about the 380K level (Figs. 5 and 16 are based on this level)? A brief statement of the sensitivity of your results to the different potential temperatures in the lower stratosphere may be useful. Also, is the MLS averaging kernel applied to the model data for a more accurate comparison to MLS data?*

    Thank you for this questions. We preferred to consider the H2O distribution at 400K, as the 380K surface may be well located below the tropopause in certain regions *e.g. Asian monsoon) and hence mixes stratospheric and tropospheric characteristic. However, for model and reanalysis intercomparison the level 380K is often used. To have better comparability to these studies, we changed Fig.16 380K now.

    For MLS data the averaging kernel was not applied. As shown by Ploeger et al. (2013) the MLS averaging kernel has only a weak effect on H2O at middle and low latitudes in the lower stratosphere, while having a strong effect at high latitudes. As the main focus of this paper is an intercomparison of different model simulations, we prefer to show the full model resolution without smearing out structures with the averaging kernel (p.11, L5).

    "... Oscillations in MLS $H_2O$ at high latitudes are a known effect of the broad averaging kernel (Ploeger et al., 2013). At low latitudes the effects of the MLS averaging kernel on $H_2O$ are much smaller and we do not apply it to the model data here, in order not to smear out the structure in the simulated $H_2O$..."

12. *p8, L9-10: "The impact of horizontal transport. . ..mixing strength covered." Effects look very similar to me. I would suggest removing this statement to avoid confusion.*

We removed this statement.

13. *Figure 5: There are four temperature contours in panels a and b, but only three contour levels are mentioned in the caption. Devise a better way to label these temperature contours?*

Thank you for this remark. We now added the temperature values in Fig.5 to clarify the plot. The small contours in the Figs.5ab, without the labels belong to 193 K (what should be obvious from the plot and labelling). Also we changed the caption for Fig.5.

14. *p11, L7: Please be more specific about what you mean by "agree slightly better" (quantify).*

We changed the text in the revised version (p12, L5).

"...Overall, regarding the global $H_2O$ distributions and maps in the LS, CLaMS modelling results with ERA-Interim are drier when compared to JRA-55, resulting from lower TTL temperatures in ERA-Interim. The agreement between CLaMS based on ERA-Interim and JRA-55 with the observations strongly depends on the considered region and season. And it is not possible to conclude from our analysis which reanalysis results in simulated $H_2O$ in the best agreement with the observations..."

15. *p11, L14: I think you mean to say "... show that the interhemispheric transport is rather unimportant..."*

It is changed in the revised version.

"...the insignificant difference between the reference (REF) and an equatorial transport barrier (BAR-0) simulations shows that the interhemispheric transport is rather unimportant for tropical mean $H_2O$ mixing ratios..."

16. *p13, L11: Where is the "double peak structure"?*

We added a short explanation to the text.

"...In the tropics, the age distribution in Fig. 7b shows a weak double peak structure up to about 500 K, indicating that the subtropics are regions of particularly fast transport, likely related to subtropical processes like monsoon circulations..."

17. *Figures 12 and 13: Why are these two figures separated? It makes more sense to combine the two and make a 4 x 4 panel figure: TWV, H2O, 2CH4, AoA. Does the sum of Fig. 12a and Fig. 13b equal Fig. 12e?*

Thank you for the remark. We had the two figures combined in a first draft version. But, we think that it is confusing to have a 4 x 4 panel Figure. So, we decided to separate it in two plots. In our opinion, it makes sense to separate it in this way (H2O + TWV; and transport tracers CH4 + AoA).

18. *Figure 15: The lines in these figures are very difficult to see.*

Thank you for pointing that out. We changed the representation of Figure 15 (now it is Fig. 16).

19. *p20, top: It appears that during boreal summer, weak mixing moistens the Asian monsoon region while very strong mixing moistens the N American monsoon region. Do you have an explanation for this interesting result?*

This is indeed an interesting result. However, we have no good explanation for this hitherto. We will study his point in the future. Additionally, now we added the discussion about this issue into the text (p22, L12).

"...During boreal winter the SH subtropical jet substantially moistens with increasing mixing, during boreal summer the NH jet moistens. In particular, the moist anomaly of the Asian and American monsoons during boreal summer is affected by small-scale mixing. Without mixing only a weak anomaly occurs in the Asian monsoon, while the moist anomaly in the American monsoon is absent. With increased mixing the Asian monsoon moist anomaly first increases (MIX-weak and REF cases). When mixing becomes very strong (MIX-strong) this behaviour changes and the entire jet region becomes strongly moistened such that the Asian monsoon moist anomaly relative to the entire jet region weakens... Overall, small-scale mixing in the CLaMS simulations and related diffusive upward moisture moisture transport seem crucial for the development of Asian and American monsoon moisture anomalies, and in particular for the American monsoon (where no anomaly occurs without including small-scale mixing)..."

20. *p20, L30: Where do you get "380K"? The explanation you provide is certainly plausible, but do these biases occur in mid and high latitudes where the H2O biases are observed?*

Thank you for pointing this out. Fig.17 (now it is Fig.18) is done for mid-and high latitudes (we forgot to mention it in the caption), we changed the description in the revised version accordingly. The related discussion in the revised version is improved (p25, L13).

"...This behaviour is the clearest at around 370 K (Fig. 18), but it is also visible at levels below and above (not shown). The different shape of the JRA-55 PDFs, with the peak at much higher mixing ratios, suggests that high $H_2O$ mixing ratios are deposited in the extratropical LS, from potential temperature levels of about 350 K up to at least about 400 K, potentially related to the convective scheme in the reanalysis...."

21. *Figure 17: It would be helpful to add MLS data to these plots.*

Thank you for the suggestion. But, we would not like to overload Fig.17 with too many lines. A comparison between CLaMS and MLS data has been made already at an earlier stage in the paper. Here we want just to discuss and compare the reanalysis with CLaMS model results (which were based on the temperature and winds taken from the same reanalysis, ERA-Interim and JRA-55). The aim is to emphasize that the JRA-55 own reanalysis $H_2O$ products have too high values and are different from CLaMS $H_2O$, which is actually consistent with JRA-55 tropopause temperatures (as it is based on a rather simple dehydration parametrization).

**Technical Corrections**

Thank you for these detailed corrections. We changed everything regarding to your suggestions.

1. *p6, L15: ". . ..our study, . . ."*

   Corrected in the revised version.

2. *p8, L7: ". . . tropical (10S-10N) entry of H2O . . ."*

   Corrected in the revised version.

3. *p8, L14: Rewrite this sentence ". . . reaches the values of âĹij0.8 ppmv" as ". . . drying of âĹij0.8 ppmv in (month)".*

   Corrected in the revised version.

4. *p10, L17: ". . . LS H2O between MLS and ClaMS simulations . . ."*

   Corrected in the revised version.
5. *p11, L13; p12, L1: I'm slightly confused by the use of the term "vanishing differ-ence". Are the differences decreasing over time?*

   Corrected in the revised version.

6. *p 12, L3: ". . . barrier at 15 N/S changes the PDF . . ."*

   Corrected in the revised version.

7. *p12, L5: ". . . (blue line in Fig 6b), . . ."*

   Corrected in the revised version.

8. *p12, L7: Rearrange the sentence: "equatorial transport barrier has no effect and cross-equatorial transport is unimportant"*

   Corrected in the revised version.

9. *p14, L9: Replace with ". . .very weak drying effect on. . ."*

   Corrected in the revised version.

10. *p16, L5: I'm not sure what you mean by "Therefore"*

    We removed that sentence, and reformulated the previous statement. Corrected in the revised version.

11. *p16, L12: " H2O (a-d) and total water (e-h)"*

    Corrected in the revised version.

12. *p20, L18: "On the contrary, . . ." is a very long sentence.*

    Corrected in the revised version.

---

## Author Comment (AC3) · 23 Mar 2018

We thank the referee for the review and for the helpful and detailed comments. We give a point-by-point reply below, where the reviewer comments are repeated in italics. The positions of the corrected sentences in the revised version are noted in the brackets, and the revised text is also given in the quotation marks point-by-point below.

[Figure]

**General remarks**

*The paper by L. Poshyvailo et al. tests the importance of three processes (tropopause temperature, horizontal transport and small-scale mixing) for lower stratospheric water vapour concentrations. The paper is overall well written and could be a valuable addition to our understanding and modelling of the tropical upper troposphere and lower stratosphere (UTLS).*

Thank you very much for this positive comment. In the revised version all comments have been taken into account, particularly we improved the discussion part, presentation of figures and formulation of the statements throughout the text due to the remarks of the Reviewer #2.

**Specific Comments**

1. PAGE 1

   *11: Are JRA-55 and ERA-Interim sufficient to constrain the space of possibilities given by current reanalysis datasets?*

   This is indeed a good remark. In our opinion, the two reanalysis datasets used are not likely sufficient to constrain the space of all possibilities. However, they are two of the three most modern and sophisticated available reanalysis (ERA-Interim, JRA-55, MERRA-2). Due to data storing capacity reasons we could not add a MERRA-2 simulation. But we think the two considered simulations in the paper provide a useful conservative limit of the space of possibilities (which can only be larger if another reanalysis is added). In particular the very new (not entirely available yet) ECMWF reanalysis ERA-5 would be another dataset which could be compared to the results of this paper. We are working on preparing such a simulation, but this is clearly beyond the scope of this paper.

*12: cancel second mention of "JRA-55 reanalysis"*

Corrected in the revised version.

*12-15: since you mention the difference due to uncertainties in tropopause temperatures (0.5 ppmv), could you provide more quantitative statements for the other two processes?*

Thank you for this remark. The statements are added in the revised version (p1, L14).

"...Comparison of tropical entry $H_2O$ from the sensitivity $15°$ N/S barrier simulation and the reference case shows differences of up to around 1 ppmv... For the sensitivity simulation with varied mixing strength differences in tropical entry $H_2O$ between the weak and strong mixing cases amount to about 1 ppmv, with small-scale mixing enhancing $H_2O$ in the LS...."

*21: reference to Nowack et al. (2015) could be added here. Nowack et al. A large ozone-circulation feedback and its implications for global warming assessments, Nature Climate Change 5, 41-45, 2015*

Corrected in the revised version (p2, L2).

*23: reference to Maycock et al. was published in Journal of Climate, see typo in reference list. Reference to Nowack et al. (2017) could be added here for another recent example of how UTLS processes/stratospheric water vapor can influence climate variability: Nowack et al. On the role of ozone feedback in the ENSO amplitude response under global warming. Geophys. Res. Lett. 44, 3858-3866, 2017.*

Thank you for these suggestions. We added the references in the revised version (p2, L2).

"...Stratospheric water vapour ($H_2O$) is a crucial factor for global radiation, as it cools the stratosphere and warms the troposphere (e.g., Forster and Shine,

1999, 2002; Shindell, 2001; Nowack et al., 2015). Particularly, changes in $H_2O$ mixing ratios in the upper troposphere and lower stratosphere (UTLS) may have significant effects on climate variability (Solomon et al., 2010; Riese et al., 2012; Maycock et al., 2013; Nowack et al., 2017)..."

2. PAGE 2

*12/13: you cite studies about the importance of the Asian Monsoon below. Would add those references here already.*

Corrected in the revised version (p2, L18).

"...Horizontal transport between the TTL and middle latitudes is strongly influenced by the Asian monsoon anticyclone and other subtropical circulation systems (e.g., Bannister et al., 2004; James et al., 2008;Wright et al., 2011; Randel and Jensen, 2013)..."

*16: to avoid confusion with too many 'lows' I recommend to change to: freeze-dried to stratospheric values, which is self-explanatory.*

Thank you for the suggestion. It is corrected in the revised version.

"...Related to the mean upward transport, the TTL includes the region of very low temperatures around the cold-point tropopause, where the moist tropospheric air is freeze-dried to stratospheric values (Brewer, 1949)..."

*19: next to Hardiman, 2015, you could add Schoeberl et al. 2014 here: Schoeberl et al. Cloud formation, convection, and stratospheric dehydration, Earth and Space Science 1, 117, 2014.*

Thank you for the remark. It is corrected in the revised version (p2, L27).

"...The freezing is sensitive not just to large-scale TTL temperatures, but also to microphysical processes controlling the ice crystal number densities, particle size distribution, and fall speed. Several studies focused on the modelling of the detailed cloud microphysical processes (e.g., Jensen and Pfister, 2004;

Jensen et al., 2005, 2012). Other recent papers have examined the effect of cloud microphysical processes on the humidity of the TTL and stratosphere using cloud models of varying complexity (e.g., Ueyama et al., 2015; Schoeberl et al., 2014)..."

3. PAGE 3

   *5: reanalysis,*

   Corrected in the revised version.

   *6: delete "largely"*

   Corrected in the revised version.

   *7-11: I agree that the lack of discussion on ozone as a process in the introduction is a shortcoming of the current manuscript. In this paragraph, there would be an opportunity to mention the importance of ozone in modulating stratospheric water vapor concentrations. In the discussion section, ozone could be included as a potential future research interest, because ozone will equally be transported by mixing etc. An ozone-focused analysis is beyond the scope of this study though, so not necessary, but a brief discussion would be useful.*

Thank you for the this suggestion. Although this paper focusses on processes controlling stratospheric $H_2O$, we added a brief discussion on stratospheric ozone (p3, L21).

"... Furthermore, it has been pointed out that the coupling between ozone, the tropospheric circulation, and climate variability plays an important role in climate change (Nowack et al., 2017). Recent studies have shown that stratospheric ozone changes may cause an increase in global mean surface warming, mostly induced by changes in long-wave radiative feedbacks due to the tropical LS ozone and related stratospheric $H_2O$ and cirrus cloud changes (e.g., Nowack et al., 2015; Dietmuller et al., 2014). Seasonal variations of LS ozone lead to a magnification of the seasonal temperature cycle in the tropics (Fueglistaler et al., 2011).

Investigation of these additional effects of stratospheric ozone is an important topic of future research focussed on stratospheric $H_2O$ feedback..."

*25: Based on model simulations,*

Corrected in the revised version.

*26: can cause*

Corrected in the revised version.

"...in the large-scale flow, can cause strong effects on..."

*33: change to: "with respect to two meteorological datasets...", following sentences add references to the recent studies on reanalysis data indicated above.*

Thank you for the remark. We changed this sentence accordingly and added more detailed explanation in the discussion (p4, L6).

"...In this paper, we investigate uncertainties of modelling $H_2O$ in the LS with respect to two meteorological datasets, ERA-Interim and JRA-55 (e.g., Dee et al., 2011; Kobayashi et al., 2015; Manney et al., 2017; Davis et al., 2017; Manney and Hegglin, 2018), used to drive transport and freeze-drying, horizontal transport between tropics and extratropics, and small-scale mixing in the Chemical Lagrangian Model of the Stratosphere (CLaMS)..."

4. PAGE 4

*27: about 2 million*

Corrected in the revised version.

*31: given the importance of the parameter for the simulations here, a somewhat more detailed description would be helpful rather than just referring to other papers.*

Thank you for this suggestion. But we think, that the short description as presented in the paper provides all the necessary information. This parameter (Lyapunov exponent) controls the strength of the mixing by defining the critical radius for merging or inserting the new air parcels in CLaMS. The further details about the CLaMS mixing parametrization are presented in the cited paper and is accessible for the reader.

5. PAGE 5

   *5: the naming convention here implies that this pcold_point is the ambient pressure in the tropopause, but this does not seem to fit your explanation? Either be more specific about the ambient pressure or change the naming of the parameter.*

   Thank you for this comment regarding clarity of the description. We changed the wording here (p5, L33).

   "...The lower boundary for $H_2O$ in CLaMS is taken from reanalysis (ERA-Interim or JRA-55) specific humidity below about 500 hPa. If saturation along a CLaMS air parcel trajectory exceeds a critical saturation (100% with respect to ice), then the $H_2O$ amount in excess is instantaneously transformed to the ice phase and partly sediments out. Such simple parametrisation has been adopted in several global Lagrangian studies (e.g., Kremser et al., 2009; Stenke et al., 2008). The saturation mixing ratio is calculated as $\chi_{H_2O} = p_s/p$ for each air parcel trajectory, with the saturation pressure given by $p_s = 10^{-2663.5/T+12.537}$ (Marti and Mauersberger, 1993), where $p$ is the ambient pressure (e.g., Kremser et al., 2009)..."

6. PAGE 6

7. PAGE 7

   *3-5: in Table 1, the no mixing experiment is labelled with a mixing parameter of 0, whereas here you say that larger values represent weaker mixing. How does this fit together?*

   Thank you for this remark. We agree that the description was misleading. Zero is the value of the critical Lyapunov exponent which should be chosen (technically)

[Figure]

in CLaMS to turn off the mixing. In theory, this corresponds then to a critical Lyapunov exponent of infinity. We clarified the text by using a Lyapunov exponent of infinity for the no mixing case in the revised manuscript (p7, L15).

"...Furthermore, we carry out a simulation without small-scale mixing (mixing in CLaMS was switched off), equivalent to a critical Lyapunov exponent of infinity..."

*5/6: it is however unclear how non-linear the effects of mixing scale with the mixing strength. This should at least be pointed out, or alternatively could be tested by running additional simulations with intermediate size parameters.*

Thank you for the suggestions. We added the information about connection between small-scale mixing and vertical diffusivity coefficient in the revised manuscript and added a new Fig. 19 (p.25-27).

"...In addition, we estimated the vertical diffusivity coefficient for the TTL for the different model simulations. The result suggests a non-linear response of H2O to the small-scale mixing in CLaMS (details are considered in Sect. 4)..."

"...Although, it is clear qualitatively that a decreasing critical Lyapunov exponent enhances mixing, it is also desirable, at least for comparison purposes, to quantify this effect. Because of the similarity between the mixing procedure in CLaMS and physical diffusion, the vertical mixing intensity can be quantified by computing the induced vertical diffusivity $K_z$ (in m$^2$/s) (Konopka et al., 2007)... Finally, it should be noted that the mixing in CLaMS induces both vertical and horizontal diffusion. However, given the larger vertical gradients of H$_2$O compared to horizontal gradients in the UTLS, the impact of small-scale horizontal diffusion is assumed to be much smaller than the impact of vertical diffusion, especially in the tropics..."

8. PAGE 8

*13/14: These values should also be given above, especially in the abstract.*

Thank you for the remark. And we added these values to the abstract (p1, L14), (p9, L4).

"...Comparison of tropical entry $H_2O$ from the sensitivity $15°$ N/S barrier simulation and the reference case shows differences of up to around 1 ppmv... For the sensitivity simulation with varied mixing strength differences in tropical entry $H_2O$ between the weak and strong mixing cases amount to about 1 ppmv, with small-scale mixing enhancing $H_2O$ in the LS..."

"...Figure 2 shows the annual cycle of tropical ($10°$S-$10°$N) stratospheric entry $H_2O$ at 400 K for all simulations. While a clear annual cycle is evident for all cases, the mixing ratios vary by more than 1 ppmv between the simulations... Suppressing horizontal transport from the subtropics into the tropics (BAR-15) significantly dries the tropical entry $H_2O$, with difference to the reference of up to around 1 ppmv. For the mixing sensitivity simulations, the largest difference from the reference case occurs for the case without mixing (MIX-no), with the MIX-no simulation dryer by about $\approx$ 0.8 ppmv in September-October..."

*Figure 2: x-axis, label DJFM..., or Dec Mar...currently it is unclear if you start in January? December?*

Thank you for this comment. We changed Figure 2 accordingly.

*15: good opportunity to cite other studies that have looked at this before. Any differences?*

Thank you for this comment. We added a reference to the discussion to the text here, and also this topic is shortly discussed in the discussion (p9, L10).

"...The CLaMS simulation driven with JRA-55 shows moister values in the TTL compared to the ERA-Interim simulation, in agreement with the recent findings of Davis et al. (2017) (for details see Sect. 4)..."

9. PAGE 9

*Figure 3: Remove space between the second and third month of each seasons label. I think it would also be better to plot the MLS climatology in the first row and differences relative to that in the lower three rows. This would make the structure of the differences more obvious.*

Thank you for the remark. Actually, there is no space between the second and third month of each seasonŽs label. Plotting the difference is a good idea, but we prefer to plot the full values to see the proper H2O distribution.

10. PAGE 10

   *2: no comma after detail*

   Corrected in the revised version.

   "...They will be investigated in more detail in the following..."

   *10: either...or*

   Corrected in the revised version.

   "...Comparison between the two simulations, driven by either ERA-Interim or JRA-55 (third and fourth row)...

11. PAGE 11

   *Figure 5: revise titles of the subfigures. Again, I would recommend plotting differences for (b), (c), (e) and (f), especially given that the rainbow color scale skews the perspective on the maps, while the color choice is also not ideal (colorblind readers). I recommend choosing a different color scale.*

   Thank you for the remark. We tried different colour schemes, but this one highlights the main features of the H2O distribution in the best way. Also we added the values of the temperature to the temperature contours for better understanding of the Figures.

[Figure]

12. PAGE 12

   *5: skewed towards...*

   Corrected in the revised version.

   "...PDF appears more strongly skewed towards low mixing ratios...

   *Fig. 6(a)-(c) should be reordered to match the order of the discussion in the text, or at least NH and SH swapped.*

   Thank you for the comment. Here we changed the order of the description, from Fig.6a to Fig.6c (p12, L13 - p14, L22), and we added an explanation regarding Fig.6b.

13. PAGE 13

   *Figure 7: again I would replace the color scale by something gradual, e.g. cold-to-warm color scale. Currently, the differences are really hard to see. Plotting differences in (b) and (c) would help, too. Maybe use contour lines for REF in order to use a single color scale?*

   Thank you for pointing this out. We agree, that the Figure 7 was not clear for representing the differences between the sensitivity studies. We changed the Fig.7, where we also show the differences between REF and BAR-35 CLaMS simulations. The results agrees well with previous studying of Garny et. al. (2014). We also added more explanation regarding this Figure (p15, L1).

   "...The pure transport effects of horizontal exchange between tropics and mid-latitudes are evident from mean age of air (AoA), the mean transit time for air through the stratosphere for the different model experiments with horizontal transport barriers. Figure 7 shows CLaMS calculations of the AoA for the reference case (Fig. 7a), simulation with transport barriers in the subtropics at 35° N/S (Fig. 7b) and the absolute difference between them (Fig. 7c). These horizontal transport barriers at 35° N/ effectively isolate the tropical pipe from the in-mixing

of older stratospheric air from mid-latitudes... A similar result has been recently found by Garny et al. (2014). Furthermore, Garny et al. (2014) presents a nice explanation of the recirculation process, describing recirculation as a process when an air parcel enters the tropical stratosphere and travels along the residual circulation to the extratropics, where it can be mixed back into the tropics, and thus recirculates along the residual circulation again. In this way, the age of air of the parcels increases steadily while performing multiple circuits through the stratosphere..."

*7: typo reference*

Thank you for this remark. It is corrected in the revised version.

*14: Why is there a consequent increase in age of air in the global stratosphere if the age is increased in the extratropics?*

Thank you for pointing this out. We removed the discussion about the "global stratosphere", as the AoA indeed is increased in the lower stratospheric extratropics.

"...therefore significantly increases AoA in the extratropical stratosphere..."

14. PAGE 14

*Figure 8: again difference plots relative to REF would be better, same changes as above concerning the color scale.*

Thank you for this remark. We added an extra plot with the largest differences between the CLaMS sensitivities and the reference simulations (Fig. 9). BAR-0 and BAR-15S showed negligible difference with REF simulation, so we do not show them here.

*10: simulations typo*

Corrected in the revised version.

15. PAGE 15

16. PAGE 16

17. PAGE 17

18. PAGE 18

*9 typo*

Corrected in the revised version.

*Figure 14: Clarify that these are percentage differences in the figure caption*

Thank you for this remark. It is corrected in the revised version (now it is Fig. 15).

19. PAGE 19

*Figure 15: again, I would suggest a change of color scale, the plots are very hard to read. In addition, change (b,c,d,f,g,h) to difference plots relative to MIX-no. Too much information for a single plot due to the two types of contour lines.*

Thank you for the remark. Plotting the differences is a good idea. However, we prefer to show the full values to present the proper $H_2O$ distribution. We also changed here the colour schemes to highlight the main features of $H_2O$ in the best way. To have better comparability between the studies in our paper, we plot this Figure now at 380K, not at 400K as it was before (it is Fig.16 now).

20. PAGE 20

*10: reliably well, or maybe fairly well or just wellÂĂÂŹ?*

Thank you for this remark. It is corrected in the revised version.

21. PAGE 21

*Figure 16: revise titles*

We do not understand what should be changed in the titles. We think the titles are clear and include all necessary information.

22. PAGE 22

*Figure 17 (and the corresponding text): I agree with Reviewer 1 that this part of the reanalysis discussion could be kept much shorter. Issues around stratospheric water vapor are known for reanalysis datasets as the authors point themselves out in the text. Discussing this topic again here in so much detail and with an extra figure distracts from the key messages of the paper, i.e. how the three sub-processes influence stratospheric water vapor concentrations.*

Thank you for the comment. We shortened a bit the description of Fig.17 (now it is Fig. 18). In our opinion the discussion is resumed enough now. Additionally, we added a discussion about recent papers on the reanalysis datasets.

"...Recent studies have emphasised the overall qualitatively very good agreement between the large-scale climatological features in the UTLS in different reanalysis datasets, while important quantitative differences remain (e.g., Manney et al., 2017). This qualitative agreement among the reanalysis in many regions of the UTLS and different seasons points to the robustness of the representation of related transport and chemistry in the reanalysis datasets (Manney and Hegglin, 2018). As the stratospheric $H_2O$ in the reanalysis is not assimilated directly, the treatment of $H_2O$ in the particular reanalysis product plays an important role. For instance, JRA-55 does not contain a parametrisation of methane oxidation in contrary to ERA-Interim (Davis et al., 2017). Davis et al. (2017) further showed that the JRA-55 mean $H_2O$ values are much too large at 100 hPa. Our results of excessively high $H_2O$ values in the JRA-55 data product in the extratropical LS agree well with the findings of Davis et al. (2017). Furthermore, Davis et al. (2017) points out that there is still a lack of assimilated observations and that significant uncertainties remain in the representation of the relevant physical processes in the reanalysis..."

---

## Referee Report (RR1)

Review of the revised manuscript "Sensitivities of modelled water vapor in the lower stratosphere: temperature uncertainty, effects of horizontal transport and small-scale mixing" by Liubov Poshyvailo et al.

The authors addressed the comments very well and the manuscript now reads well. The expanded analysis on vertical diffusivity is also interesting. I recommend publication after addressing the following very minor suggestions and technical errors.

1.  Should there be a distinction between the terminology "tropical entry H2O" and "stratospheric H2O"? The two appears to be used interchangeably, but they refer to slightly different quantities so I found myself trying to figure out which. For example, the Abstract notes "difference of up to around 1 ppmv" in tropical entry H2O due to transport across the 15NS barrier (also mentioned in the Conclusions section). The description of Figure 9a refers to the 1ppmv impact on stratospheric H2O. Figure 6b seems to show a mean difference of ~0.5 ppmv in the domain 20S-20N, 380-420K, which is essentially the tropical entry H2O.
2.  Are the temperature, zonal wind and pressure level contours in Figure 3 derived from ERA-Interim reanalysis? I believe they are the same across the four (DJF and JJA) panels, correct?
3.  It looks as though Figure 16 shows potential vorticity contours in black (not grey) and temperature contours in white (not violet).
4.  In Section 4.2 (which I like), the authors mention the study by Ueyama et al. (2015) who varied the vertical diffusivity coefficient by 2 (not 3) orders of magnitude (from 0.001 to 0.1 m2/s) and found a moistening effect of ~0.5 ppmv.